# Computational modeling of threat learning reveals links with anxiety and neuroanatomy in humans

**Rany Abend**[1]*[†], **Diana Burk**[2][†], **Sonia G Ruiz**[1], **Andrea L Gold**[1,3], **Julia L Napoli**[2], **Jennifer C Britton**[1,4], **Kalina J Michalska**[1,5], **Tomer Shechner**[1,6], **Anderson M Winkler**[1], **Ellen Leibenluft**[1], **Daniel S Pine**[1], **Bruno B Averbeck**[2]

[1]Emotion and Development Branch, National Institute of Mental Health, National Institutes of Health, Bethesda, United States; [2]Laboratory of Neuropsychology, National Institute of Mental Health, National Institutes of Health, Bethesda, United States; [3]Department of Psychiatry and Human Behavior, Brown University Warren Alpert Medical School, Providence, United States; [4]Department of Psychology, University of Miami, Coral Gables, United States; [5]Department of Psychology, University of California, Riverside, Riverside, United States; [6]Psychology Department, University of Haifa, Haifa, Israel

**\*For correspondence:**
rany.abend@nih.gov

[†]These authors contributed equally to this work

**Competing interest:** The authors declare that no competing interests exist.

**Abstract** Influential theories implicate variations in the mechanisms supporting threat learning in the severity of anxiety symptoms. We use computational models of associative learning in conjunction with structural imaging to explicate links among the mechanisms underlying threat learning, their neuroanatomical substrates, and anxiety severity in humans. We recorded skin-conductance data during a threat-learning task from individuals with and without anxiety disorders (N=251; 8-50 years; 116 females). Reinforcement-learning model variants quantified processes hypothesized to relate to anxiety: threat conditioning, threat generalization, safety learning, and threat extinction. We identified the best-fitting models for these processes and tested associations among latent learning parameters, whole-brain anatomy, and anxiety severity. Results indicate that greater anxiety severity related specifically to slower safety learning and slower extinction of response to safe stimuli. Nucleus accumbens gray-matter volume moderated learning-anxiety associations. Using a modeling approach, we identify computational mechanisms linking threat learning and anxiety severity and their neuroanatomical substrates.

## Editor's evaluation

The authors present an investigation into the role of threat learning processes in symptoms of anxiety across a broad sample of subjects with and without clinical anxiety, across multiple age groups. Authors demonstrated weaker safety learning in those who were more anxious.

## Introduction

Threat learning encompasses processes that rapidly generate associations between neutral stimuli and aversive outcomes. Influential theories implicate variations in the mechanisms underlying such conserved associative-learning processes in anxiety symptoms (*Mineka and Oehlberg, 2008*; *Duits et al., 2015*). Computational learning theory offers tools to quantify associative learning dynamics as they relate to variations in these mechanisms. Here, we study physiological data recorded during a threat learning paradigm in a large sample (N = 251) featuring a wide range of normative to

pathological anxiety severity across childhood to adulthood. We use these data to uncover links among threat learning processes, their neuroanatomical substrates, and anxiety severity via latent learning parameters estimated by reinforcement learning models.

Neural circuits have evolved to allow rapid learning of threat associations following encounters with danger (*Fanselow, 2018*; *LeDoux, 2014*; *LeDoux, 2000*). Through threat conditioning, a neutral stimulus rapidly comes to elicit fear responding in anticipation of danger (*Fanselow, 2018*). Through extinction, such conditioned anticipatory responding is attenuated if the stimulus no longer predicts the occurrence of threat. Extensive research across species implicates conserved neural circuitry in these processes, highlighting their adaptive value (*LeDoux, 2000*).

At the same time, aspects of these processes could prove maladaptive. Thus, influential theories link individual differences in threat learning to the emergence and persistence of excessive threat-anticipatory fear responses which are central in expression of anxiety symptoms (*Mineka and Oehlberg, 2008*; *Duits et al., 2015*; *Lissek, 2005*; *Barlow, 2002*; *Abend et al., 2021*; *Corchs and Schiller, 2019*). Moreover, aberrant maturational processes in the circuitry supporting such learning may underlie the emergence of anxiety symptoms in childhood and adolescence (*Beesdo et al., 2009*; *Kessler et al., 2012*; *Casey et al., 2015*; *Pattwell et al., 2012*; *Lau et al., 2011*; *Craske et al., 2018*; *Craske et al., 2012*). However, studies in youth and adults attempting to link threat learning and anxiety symptoms yield inconsistent findings (*Duits et al., 2015*; *Lissek, 2005*; *Dvir et al., 2019*), hindering neuroscience research on normal and abnormal threat learning mechanisms.

Failure to detect robust and replicable associations between threat learning and anxiety symptoms could reflect limitations of standard analytic methods because these are not designed to capture associative learning dynamics (*Lonsdorf et al., 2017*; *Ney et al., 2018*). These limitations could potentially be overcome through applications of computational learning theory, which provides a mathematical framework for quantifying the temporal dynamics of associative learning processes, as indexed with latent variables (*Garrison et al., 2013*; *Li and McNally, 2014*; *Keiflin and Janak, 2015*; *Rescorla and Wagner, 1972*). Based on the measurement of prediction errors between expected and experienced outcomes, associative learning models were initially applied to reward learning (*Keiflin and Janak, 2015*; *Schultz, 2013*; *Lee et al., 2012*), including examining reward learning perturbation in anxiety (*Huang et al., 2017*; *Aylward et al., 2019*). Recent studies extend this approach to model associations between threat cues and aversive outcomes. These include studies in healthy participants (*Wise et al., 2019*; *Tzovara et al., 2018*; *Zhang et al., 2016*; *Li et al., 2011*; *Schiller et al., 2008*; *Atlas et al., 2016*), individuals with elevated anxiety symptoms (*Browning et al., 2015*; *Michalska et al., 2019*), and in adult patients with post-traumatic stress disorder and transdiagnostic symptom dimensions (*Homan et al., 2019*; *Gagne et al., 2020*; *Wise and Dolan, 2020*). These studies utilize variations of threat learning paradigms, such as reversal or instrumental conditioning procedures, to identify parameters that determine important aspects of learning, including some that relate to psychopathology. Here, we complement this body of work by studying the mechanisms that link individual differences in rapid threat conditioning and extinction to anxiety severity across the lifespan (*Mineka and Oehlberg, 2008*; *Duits et al., 2015*; *Lissek, 2005*).

In the current report, we address this gap by applying reinforcement learning models to skin conductance response (SCR) data recorded during a threat learning paradigm, in conjunction with structural brain imaging, to quantify age-dependent associations among threat learning processes, neuroanatomy, and anxiety severity. We complement and extend prior work in several ways. First, we study a sample that includes individuals with and without anxiety disorders falling along a wide age span, from age 8 to 50 years. This sample contains wide ranges of anxiety symptoms and age, which increase statistical power to detect associations among age, anxiety, and threat learning indices. Second, previous work modeled cue responding over many reinforcement trials (e.g. > 60 trials) (*Homan et al., 2019*; *Tzovara et al., 2018*; *Zhang et al., 2016*), which may more closely model the *expression* of conditioned responses. Here, we focus instead on the rapid *learning* of threat associations that takes place during a shorter schedule (*Fanselow, 2018*). Third, we model four learning effects: threat conditioning, threat generalization, safety learning, and threat extinction. This modeling approach provides tests of different theories that link mechanistic variations in these specific processes to the severity of anxiety (*Mineka and Oehlberg, 2008*; *Lonsdorf et al., 2017*; *Craske et al., 2012*; *Abend et al., 2020*; *Michalska et al., 2017*; *Curzon et al., 2009*). Finally, we use data from a relatively large sample (N = 215) of medication-free subjects, enabling parameter estimation

with greater accuracy and over a large model space. We hypothesize that anxiety severity is associated with latent learning parameters (*Mineka and Oehlberg, 2008*; *Duits et al., 2015*); that age moderates specifically the association between extinction learning and anxiety (*Casey et al., 2015*); and that morphometry features in subcortical structures and prefrontal cortex moderate associations between learning parameters and anxiety (*Fullana et al., 2018*; *Maren, 2001*; *Fullana et al., 2016*).

## Materials and methods

### Participants

All participants were recruited from the community as paid volunteers for research at the National Institute of Mental Health (NIMH), Bethesda, MD. Written informed consent was obtained from adult participants (≥18 years) as well as parents, and written assent was obtained from youth. Procedures were approved by the NIMH Institutional Review Board (protocol 01-M-0192). Data from a sample of 351 individuals were initially considered. Due to our focus on model fitting with trial-by-trial data and the noisy nature of SCR data, we excluded individuals with excessive missing data, with similar exclusion proportion to prior work (*Homan et al., 2019*) (n=136). This led to a final sample of N=215, which included 104 healthy participants (53 females; ages 8-44 years) and 111 medication-free participants with anxiety disorders (63 females; ages 8-50 years). Healthy and anxiety groups did not differ in age, t(213)=0.97, $p$=0.774, d=0.13, sex, $\chi^2_{(1)}$=3.56, $p$=0.060, V<0.01; or IQ, t(213)=0.15, $p$=0.882, d=0.02. Analyses on raw psychophysiology data and neuroanatomy data from portions of this sample have been previously reported (*Abend et al., 2020*; *Shechner et al., 2015*; *Britton et al., 2013*; *Gold et al., 2020*). The current study reports novel analyses of these data. Data from 32 participants were excluded due to aborting the task (22 anxious, 8 healthy) or technical problems (1 anxious, 1 healthy). Data from four additional participants (2 anxious, 2 healthy) were excluded from analyses since they inquired and were then informed of the CS contingencies prior to the conditioning phase (*Mechias et al., 2010*).

### Anxiety severity

All participants were interviewed by trained clinicians using semi-structured clinical interviews (*Kaufman et al., 1997*; *First et al., 2002*); see Appendix 1. Anxious patients were required to meet criteria for generalized, social, and/or separation anxiety disorder. Healthy participants did not meet criteria for any psychiatric diagnosis. To assess current anxiety symptom severity, youth and adults completed standard self-report anxiety questionnaires. Youth (age < 18 years) and parents completed the Screen for Child Anxiety Related Emotional Disorders (SCARED) (*Birmaher et al., 1997*), and adults (age ≥ 18 years) completed the trait subscale of the State-Trait Anxiety Inventory (STAI) (*Spielberger et al., 1970*), within 3 months of the task. The SCARED is a child- and parent-report measure comprising 41 items assessing recent anxiety symptoms and possessing strong psychometric properties (*Birmaher et al., 1997*; *Birmaher et al., 1999*); to reduce informant differences, child- and parent-report scores were averaged (*Behrens et al., 2019*; *Abend et al., 2021*). The STAI (*Spielberger et al., 1970*) consists of 20 items relating to general anxious moods and possesses strong psychometric properties (*Elwood et al., 2012*). To combine these anxiety measures, we Z-transformed each measure within its respective age sample; these Z-scores were then combined across samples and used in analyses (*Abend et al., 2019*). These anxiety severity scores manifested a unimodal, continuous distribution across the sample (see *Figure 1—figure supplement 1*). Given that the SCARED and STAI might not capture the construct of anxiety in an identical manner, we also report on SCARED analyses in youth and STAI analyses in adult participants separately.

### Threat conditioning and extinction task

The task involved rapid, uninstructed Pavlovian conditioning and extinction of threat associations (*Fanselow, 2018*). The task has previously been found effective (i.e. produce conditioning while maintaining a low dropout rate) among individuals with anxiety and healthy participants from both youth and adult populations (*Lau et al., 2011*; *Michalska et al., 2017*; *Shechner et al., 2015*; *Britton et al., 2013*; *Gold et al., 2020*; *Den et al., 2015*; *Lau et al., 2008*; *Ryan et al., 2019*). The task consisted of a pre-conditioning phase, a conditioning phase, and an extinction phase (*Figure 1*). Photographs of two women displaying neutral expressions (*Tottenham et al., 2009*) served as the conditioned threat and

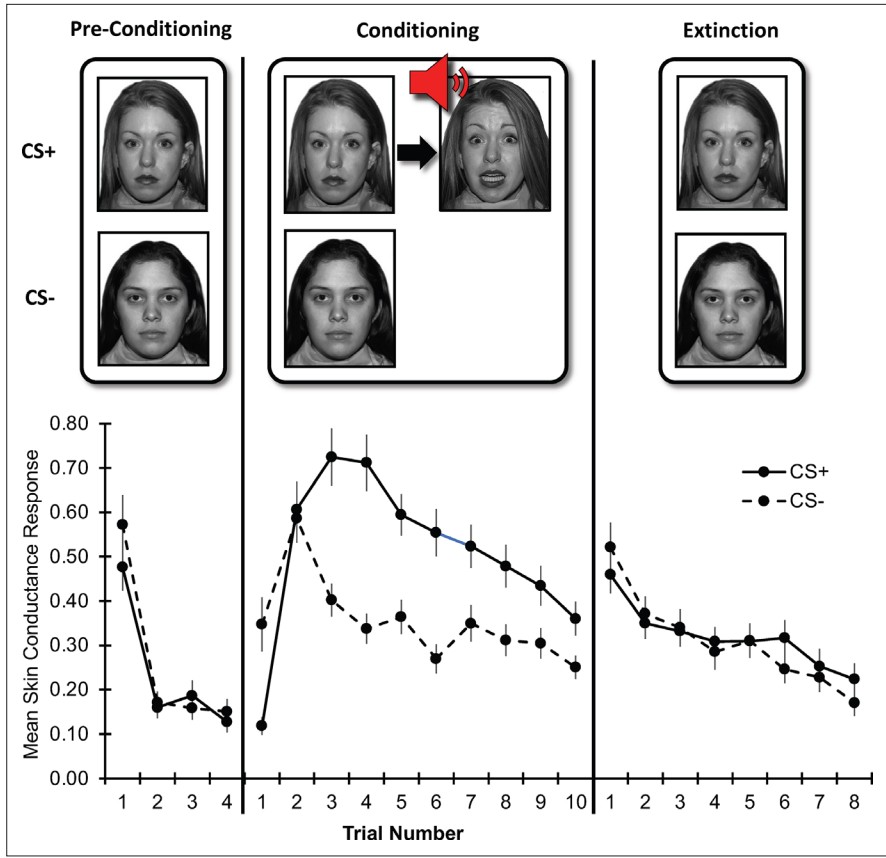

**Figure 1.** Threat learning task and physiological data. Top: Schematic representation of the threat learning paradigm. During the pre-conditioning phase, the designated threat (CS+) and safety (CS-) stimuli were presented without reinforcement. During the conditioning phase, the CS + was paired with a fearful face co-terminating with a scream (UCS); the CS- was never reinforced. During the extinction phase, both CS + and CS- were not reinforced by the UCS. Bottom: Mean raw skin conductance response for the CS + and CS- by task phase and trial. *Note:* Trial number indicates the nth trial for that stimulus. However, the CS+ and CS- trials were presented in counterbalanced order throughout the task. CS = conditioned stimulus; UCS = unconditioned stimulus. Error bars indicate one standard error of the mean.

The online version of this article includes the following figure supplement(s) for figure 1:

**Figure supplement 1.** Histogram depicting the distribution of the standardized anxiety severity scores across the sample.

safety stimuli (CS + and CS-, respectively), each presented for 7s. During the *pre-conditioning* phase, each CS was presented four times to allow physiological responses to the novel stimuli to habituate. During the threat *conditioning* phase, each CS was presented 10 times, and the CS + was followed by the unconditioned stimulus (UCS), a 1s presentation of the same actress displaying fear and co-occurring with a 95dB female scream delivered via headphones, with an 80% reinforcement schedule. Participants were instructed that they could learn to predict when the UCS would occur, but they were not explicitly informed of this contingency. During the *extinction* phase, the CSs were each presented eight times in the absence of the UCS. In all phases, the duration of the CS + and CS- presentation was 8s, but shortened to 7s when UCS occurred. An inter-trial interval (a gray screen presented for 8-21s, averaging 15s) separated the trials. Presentation order of the CSs was pseudo-randomized (two different orders counterbalanced across participants). The task was programmed and administered using PsyLab psychophysiological recording system (PsyLab SAM System, Contact Precision Instruments, London). Skin conductance, electromyography, and electrocardiography were recorded. Due to our focus on skin conductance responses as indexing conditioned anticipatory threat responses (*Lonsdorf et al., 2017*), the other measures are not analyzed in the current report.

## Skin conductance

Skin conductance was recorded at 1000Hz using PsyLab from two Ag/AgCl electrodes from the medial phalanx of the middle and ring non-dominant-hand fingers. In line with prior research, skin conductance response (SCR) was determined by the square-root-transformed difference between base-to-peak amplitude within 1-5s after stimulus onset (*Zhang et al., 2016*; *Li et al., 2011*; *Schiller et al., 2008*; *Shechner et al., 2015*; *Marin et al., 2017*; *Marin et al., 2020*). To generate more accurate estimates from trial-by-trial data, we next cleaned trial-level data. Missing skin conductance trial values and within-subject outliers were identified per subject, for each CS type during conditioning and extinction phase separately; outliers were defined and up to three consecutive values were linearly interpolated using the *zoo* package in R (*Zeileis and Grothendieck, 2005*). Next, trailing missing trial values during conditioning and leading missing values during extinction were linearly extrapolated. Subjects with more than 50% missing trial values were excluded. Then, overall CS+ and CS- SCRs were averaged for each subject. Group outliers for each CS type were identified and excluded. Based on these conservative criteria, skin conductance data from 136 participants were excluded from modeling (52 anxious, 84 healthy). This proportion is similar to excluded data in prior work (*Homan et al., 2019*; *Lonsdorf et al., 2019*). Included and excluded participants did not differ in anxiety severity (by diagnosis or continuously) or IQ; excluded relative to included participants had a higher mean age and a greater proportion of females were excluded (see *Appendix 1—table 1*).

## Raw SCR data

Raw SCR data by trial and task phase are depicted in *Figure 1*. During the conditioning phase, CS+ threat conditioning (acquisition of conditioned fear response) was noted; response to the CS- was characterized by generalization of conditioned fear response (increased response despite no reinforcement) and safety learning (diminishing response through continued non-reinforcement). During the extinction phase, rapid extinction (diminishing response) was noted for both stimuli. Statistics on these data are reported in the Results section. As noted, perturbations in these processes have been suggested to contribute to the emergence and persistence of anxiety symptoms (*Mineka and Oehlberg, 2008*; *Duits et al., 2015*; *Lissek, 2005*; *Dymond et al., 2015*; *Pittig et al., 2016*; *Vervliet et al., 2013*.) Specifically, facilitated threat conditioning (*Orr et al., 2000*), greater threat generalization (*Dymond et al., 2015*; *Vervliet et al., 2013*; *Lissek et al., 2014*), and slower safety learning (*Craske et al., 2018*; *Craske et al., 2012*) have each been theorized to lead to anxiety symptoms. Further, threat extinction processes have been implicated in both anxiety symptoms and exposure-based treatment for anxiety (*Milad and Quirk, 2012*; *Casey et al., 2015*; *Craske et al., 2018*; *Vervliet et al., 2013*). Due to the potential relevance of these learning processes to anxiety, and as they were evident in the data, we used reinforcement modeling to examine them.

## Reinforcement learning models

In previous work on these data, we used standard statistical models to assess the effects of anxiety on learning, specifically by averaging responses across trials (*Abend et al., 2020*). In this manuscript, the goal was to explore dynamic learning processes that could account for the model-agnostic effects seen in the previous study. To model these dynamic processes during threat conditioning and extinction, we fit multiple reinforcement learning models to the trial-level SCR data for each participant. The models describe the trial-by-trial changes in the associative strength between the conditioned stimulus (CS) and the aversive unconditioned stimulus (UCS). Using a family of related models, we focused on modeling specific processes that were evident in the data and have potential relevance to anxiety.

## Conditioning phase

We modeled three processes taking place during the conditioning phase: threat conditioning (*Mineka and Oehlberg, 2008*; *Orr et al., 2000*), whereby the CS+ comes to elicit a threat-anticipatory response via reinforcement by the UCS; threat generalization (*Vervliet et al., 2013*; *Lissek et al., 2014*), whereby conditioned fear response is generalized to the safe CS-; and safety learning (*Craske et al., 2018*; *Craske et al., 2012*), whereby CS- generalized fear responses are gradually diminished through non-reinforcement. We fit fourteen reinforcement learning models (variations of three main models) to CS+ and CS- conditioning data that varied in the number of parameters and features used to model the psychophysiological responses.

Twelve models were initially based on a Rescorla-Wagner (RW) model (**Rescorla and Wagner, 1972**), whereby at trial t+1, the value $v_{CS}$ of each CS is updated based on its value in the preceding trial $v_{CS}(t)$ plus the prediction error, $\delta(t)$, on trial t. We introduced several additional features to the basic model to better capture the dynamics in the data. For example, as can be seen in the conditioning phase data (**Figure 1**), participants rapidly generalized a fear response to the CS- (i.e., showed an increase in response to CS- despite non-reinforcement). A standard RW model will not capture conditioning to the CS-, as this cue was never reinforced. Generalization of the conditioned response to the CS- can be captured by a model that tracks the psychophysiological responses to the CS+ and 'projects' them onto the CS-. Therefore, our base RW model contains a CS- generalization term that updates CS- values using prediction errors from CS+ UCS trials.

The prediction error was calculated in only the CS+ trials and only when a UCS was delivered as:

$$\delta(t) = r(t) - v_{CS+}(t) \tag{1}$$

where r(t) is the SCR to the UCS on trial t. In CS+ trials the value for both the CS+ and CS- were updated according to:

$$v_{CS+}(t + 1) = v_{CS+}(t) + \alpha_{CS+}\delta(t) \tag{2}$$

$$v_{CS-}(t + 1) = v_{CS-}(t) + \alpha_{CS-}\delta(t) \tag{3}$$

We used the magnitude of response to the UCS as the reinforcement term, *r(t)*, responses to the UCS diminished across acquisition and setting *r(t)* to 1 when a UCS was delivered would not have the appropriate scale to capture SCRs to the CS; see Appendix 1 for further discussion. We also fit models (10-12, see below), in which we used an additional regression step, and in this case we used values of 0/1 for *r(t)*.

We explored other updating schemes, including crossing the updates on both CS+ and CS- trials to the other cue. However, the approach in **equations 1-3** provided the best fit. Note that below when we write the variable without subscripts, we are referring to both CS+ and CS- terms.

As noted, we explored additional features to capture learning effects, which led to fourteen total models (see **Table 1**). First, we fit a learning inertia term (**Abbott, 2008**), which assumed that the effect of the prediction error accumulated across m recent trials. Specifically, if we define the prediction error using equation 1 above, the learning inertia term updated the value for the CS+/CS- using $v_{CS}(t + 1) = v_{CS}(t) + \alpha\delta^{in}$ where:

$$\delta^{in}(t) = \sum_{k=0}^{k_m} \delta(t - k) \tag{4}$$

The value $k_m = 2$ maximized the fit across all participants. Thus, we replaced the prediction error defined in equation 1 with the prediction error defined in equation 4. This term has no free parameters at the single-subject level, as $k_m$ was adjusted once at the population level and held constant for all participants. Qualitatively, this model improved the fit, but we found only indeterminate statistical evidence to support it.

Second, we examined Bayesian learning rate decay. This assumes that the learning rate starts high and then diminishes over trials according to the information that has been accumulated (**Fiesler and Beale, 1995**). In this model variant, $\alpha_{CS+}$ and $\alpha_{CS-}$ were free parameters and the learning rate on each trial was given by:

$$\alpha(t) = \frac{\alpha}{sqrt(t)} \tag{5}$$

This variant has no additional free parameters over the base RW model.

Third, we examined habituation of the conditioned response (diminished response over trials despite reinforcement). This captures habituation in addition to decreased response to the UCS over trials. Such habituation has been observed in other studies (**Homan et al., 2019**; **Tzovara et al., 2018**), and requires an additional parameter beyond the basic RW learning parameter to account for it. Accordingly, we fit a multiplicative exponential decay term, which resulted in an SCR estimate given by:

$$v_{CS}^{hab}(t) = v_{CS}(t)e^{-\varphi_{CS}[t-t_0]^+} \tag{6}$$

**Table 1.** Specifications and estimated free parameters for each model fit to the CS- and CS+.

| Model | Model specification | Free parameters | Initialization values |
|---|---|---|---|
| 1.RW | $\nu_{CS}(t+1) = \nu_{CS}(t) + \alpha\delta$ | $\alpha_{CS+}, \alpha_{CS-},$ | $\nu_{CS+}(0) = V_i$ $\nu_{CS-}(0) = V_i$ where $v_i$ is the SCR value from the last habituation trial (acquisition) and SCR value from the first extinction trial (extinction) |
| 2.RW with inertia | $\nu_{CS}(t+1) = \nu_{CS}(t) + \alpha(t)\delta^{in}$, whereby, $\delta^{in}(t) = \sum_{k=0}^{k_m} \delta(t-k)$, with $m$=number of recent trials | $\alpha_{CS+}, \alpha_{CS-}$ | same as model 1 |
| 3.RW with Bayesian learning-rate decay | $\nu_{CS}(t+1) = \nu_{CS}(t) + \alpha(t)\delta$, with $\alpha(t) = \frac{\alpha}{sqrt(t)}$ | $\alpha_{CS+}, \alpha_{CS-}$ | same as model 1 |
| 4.RW with habituation | $v_{CS}(t+1) = v_{CS}(t) + \alpha\delta$, with $\nu_{CS}(t)$ multiplied by $e^{-\varphi_{CS}[t-t_0]^+}$ after update | $\alpha_{CS+}, \alpha_{CS-}; \varphi_{CS+}, \varphi_{CS-}$ | same as model 1 |
| 5.RW with inertia and Bayesian learning-rate decay | $v_{CS}(t+1) = v_{CS}(t) + \alpha(t)\delta^{in}$, with $\alpha(t) = \frac{\alpha}{sqrt(t)}$, and $\delta^{in}(t) = \sum_{k=0}^{k_m} \delta(t-k)$, with $m$=number of recent trials | $\alpha_{CS+}, \alpha_{CS-}$ | same as model 1 |
| 6.RW with inertia and habituation | $\nu_{CS}(t+1) = \nu_{CS}(t) + \alpha(t)\delta^{in}$, whereby $\delta^{in}(t) = \sum_{k=0}^{k_m} \delta(t-k)$, with $m$=number of recent trials, and $\nu_{CS}(t)$ multiplied by $e^{-\varphi_{CS}[t-t_0]^+}$ after update | $\alpha_{CS+}, \alpha_{CS-}; \varphi_{CS+}, \varphi_{CS-}$ | same as model 1 |
| 7.RW with Bayesian learning-rate decay and habituation | $\nu_{CS}(t+1) = \nu_{CS}(t) + \alpha\delta$, whereby $\alpha(t) = \frac{\alpha}{sqrt(t)}$ and $v_{CS}(t)$ multiplied by $e^{-\varphi[t-t_0]^+}$ after update | $\alpha_{CS+}, \alpha_{CS-}; \varphi_{CS+}, \varphi_{CS-}$ | same as model 1 |
| 8.RW with inertia and Bayesian learning-rate decay, and habituation | $\nu_{CS}(t+1) = \nu_{CS}(t) + \alpha(t)\delta^{in}$, whereby, $\alpha(t) = \frac{\alpha}{sqrt(t)}$ and $\delta^{in}(t) = \sum_{k=0}^{k_m} \delta(t-k)$, with $m$=number of recent trials, and $\nu_{CS}(t)$ multiplied by $e^{-\varphi_{CS}[t-t_0]^+}$ after update | $\alpha_{CS+}, \alpha_{CS-}; \varphi_{CS+}, \varphi_{CS-}$ | same as model 1 |
| 9.RW-PH hybrid | $\nu_{CS}(t+1) = \nu_{CS}(t) + \kappa\alpha\delta$, whereby $\alpha = \alpha(1-\gamma) + \gamma|\delta(t)|$ | $\gamma_{CS+}, \gamma_{CS-}, \kappa_{CS+}, \kappa_{CS-}$ | $\alpha_{CS+}(0) = 1$ $\alpha_{CS-}(0) = 1$ |
| 10.Hybrid(V) *Li et al., 2011* | (Changing $V_n$ for $\nu_{CS}(t+1)$ $\delta(n) = bUCS(n) - V_n(x_n)$), where $bUCS(n)$=1 if a UCS was delivered and $bUCS(n)$=0 if no UCS was delivered (b indicates binary UCS). SCR was then predicted using a regression: $pSCR(V)_n \sim N(\beta_0 + \beta_1 V_n(x_n), \sigma)$ The squared error: $Hybrid(V)Error = (pSCR(V) - SCR)^2$ | $\gamma_{CS+}, \gamma_{CS-}, \kappa_{CS+}, \kappa_{CS-}, \beta_0, \beta_1$ | $\alpha_{CS}+(0) = 1$ $\alpha_{CS}-(0) = 1$ $v_{CS}+(0) = v_i$ $v_{CS}-(0) = v_i$ where $v_i$ is the SCR value on the last habituation trial and SCR value from the first extinction trial (extinction) |
| 11.Hybrid($\alpha$) *Li et al., 2011* | Same as model 10 (changing $V_n$ for $\nu_{CS}(t+1)$) except $pSCR(\alpha_n) \sim N(\beta_0 + \beta_1\alpha_n(x_n), \sigma)$ The squared error: $Hybrid(\alpha)Error = (pSCR(\alpha) - SCR)^2$ | $\gamma_{CS+}, \gamma_{CS-}, \kappa_{CS+}, \kappa_{CS-}, \beta_0, \beta_1$ | same as model 10 |

*Table 1 continued on next page*

*Table 1 continued*

| Model | Model specification | Free parameters | Initialization values |
|---|---|---|---|
| 12.Hybrid (V+α) *Li et al., 2011* | Same as model 10 changing $V_n$ for $\nu_{CS}(t+1)$ with additional regression such that: $$pSCR(V,\alpha)n \sim N(\beta 0 + \beta 1\, Vn(xn) + \beta 2\, \alpha n(xn), \sigma)$$ $$Hybrid(V+\alpha)Error = (pSCR(V,\alpha) - SCR)^2$$ | $\gamma_{CS+}$, $\gamma_{CS-}$, $\kappa_{CS+}$, $\kappa_{CS-}$, $\beta_0$, $\beta_1$ | same as model 10 |
| 13.Mixed prior mean and uncertainty model *Tzovara et al., 2018* | $h_{CS}(t) = -ln(\alpha_{CS} + \beta_{CS})$ <br> $z_{CS}(t) = h_{CS}(t) + E[\theta]$ <br> Where $B(\alpha_{CS}, \beta_{CS})$ is a Beta function whose parameters are updated according to: <br> $\alpha_{CS}(t) = \alpha_{CS}(t-1) + u(t-1)$ <br> $\beta_{CS}(t) = \beta_{CS}(t-1) - u(t-1) + 1$ <br> Where $u(t) = 1$ if a US occurred and $u(t) = 0$ otherwise. $\beta_0$, $\beta_1$ are the regression parameters relating $z_{CS}(t)$ to SCR | $\beta_0$, $\beta_1$ | $\alpha_{CS}(0) = 1\,\beta_{CS}(0) = 1$ |
| 14.Mixed prior mean and uncertainty model (Model 13) *Tzovara et al., 2018* with habituation | Same as model 13 with the addition: $h_{CS}(t)$ multiplied by $e^{-\varphi_{CS}[t-t_0]^+}$ after update for habituation | $\beta_0$, $\beta_1$, $\varphi_{CS+}$, $\varphi_{CS-}$ | same as model 13 |

When we fit this model, we estimated $v_{CS}$ using equations 2 and 3, and then multiplied the estimated values by the habituation term given in equation 6. Note that there were two of these equations, defined by separate free parameters $\varphi_{CS+}$ and $\varphi_{CS-}$. This model, therefore, had two additional parameters for each participant which controlled the rate of habituation for the CS+ and CS- cues. The rectified linear operator, $[x]^+$, returns 0 for negative values and x for positive values. We also estimated a single value of $t_0$ across all participants, as it was clear that the habituation process did not start in the first conditioning trial. We found that $t_0 = 2$ was optimal and significantly improved the pooled model fit across participants; see Appendix 1 for additional information.

In addition to these RW models, we fit a RW-Pearce-Hall (PH) "hybrid" model to the data, given findings on its relevance to threat learning processes (*Homan et al., 2019*; *Tzovara et al., 2018*; *Zhang et al., 2016*; *Li et al., 2011*). This model utilizes *Equation 2*, *Equation 3* for the value update with absolute value of the prediction error. However, in the PH model the association parameter is adaptive, and updated on each trial as:

$$\alpha(t+1) = \gamma \mid \delta(t) \mid +(1-\gamma)\alpha(t) \tag{7}$$

And the update equation is given by:

$$\nu_{CS}(t+1) = \nu_{CS}(t) + \kappa\alpha \tag{8}$$

The variable γ is a free parameter that controls the rate of update of α, and the variable κ is a fixed learning rate. We fit separate γ and κ parameters for CS+ and CS-. Model 9 therefore, has 4 parameters ($\kappa_{CS+}$; $\kappa_{CS-}$; $\gamma_{CS+}$; $\gamma_{CS-}$).

We also fit a series of previously reported models (Models 10, 11, 12) that had an additional regression step, such that SCR was related to value, associability, or both through regression coefficients. Here, we change notation to be consistent with previous papers and write $V_n(x_n)$ instead of $\nu_{CS}$ where $x_n$ indicates CS+ and CS-, and *n* is the trial number. Model fits were conducted as previously described, except for two changes specific to these models. First, the prediction error was calculated using a 1/0 for whether a UCS was delivered or not: $\delta(n) = bUCS(n) - V(x_n)$, where $bUCS(n) = 1$ if a UCS was delivered and $bUCS(n) = 0$ if no UCS was delivered (b indicates binary UCS). Second, SCR was predicted using a regression, which can be written as a normal distribution around a mean determined by the scaled predicted value, associability, or both (*Homan et al., 2019*; *Zhang et al., 2016*; *Li et al., 2011*).

$$pSCR(V)_n \sim N(\beta_0 + \beta_1 V_n(x_n), \sigma)$$

$$pSCR(\alpha)_n \sim N(\beta_0 + \beta_1\alpha_n(x_n), \sigma)$$

$$pSCR(V, \alpha)_n \sim N(\beta_0 + \beta_1 V_n(x_n) + \beta_2\alpha_n(x_n), \sigma)$$

where, in this case, *n* is the trial number. The squared error between the scaled predicted value, associability or both was then calculated as:

$$Hybrid(V)Error = (pSCR(V) - SCR)^2$$

$$Hybrid(\alpha)Error = (pSCR(\alpha) - SCR)^2$$

$$Hybrid(V + \alpha)Error = (pSCR(V, \alpha) - SCR)^2$$

Thus Models 10 and 11 had two additional parameters and Model 12 had three additional parameters to Model 9, that is, the β regression parameters, which were not used in other models.

Finally, we adapted the (*Tzovara et al., 2018*) model, which uses uncertainty and value to co-determine the model's predictions. We modeled the response to both the CS+ and CS- jointly and used two parameters for the regression values relating uncertainty and value to the SCR (see *Table 1*). For model 14, we added an additional parameter for each type of reinforced data (CS+ and CS-) to account for the habituation observed, for a total of four parameters.

Together, we fit a total of fourteen models (see *Table 1*) to the acquisition data, where model 1 was the base RW model; model 2 included a prediction-error inertia term; model 3 assumed a learning-rate decay; model 4 included a habituation term; model 5 included the inertia term and learning-rate decay; model 6 included inertia and habituation terms; model 7 included learning-rate decay and habituation; model 8 included inertia, learning-rate decay, and habituation; model 9 was the RW-PH hybrid model; model 10 was the Hybrid(V) model; model 11 was the Hybrid(α) model; model 12 was the Hybrid(V+α) model; model 13 was the uncertainty model and model 14 was the uncertainty model with habituation. In summary, for models 1-9, predictions were not scaled to the data by using regression. Models 1-9 predict CS-related SCR from US-related SCR. For models 10-14, CS-related SCR is predicted using a regression to associability, value, or uncertainty, as described for each model. For models 1-9, outcome was modeled as a continuous variable. For models 10-14, we modeled outcome as a binary value of 0 or 1, as described in past publications using these models. For the equations for each model, see *Table 1*.

Of note, the learning processes studied here manifest over few trials, which could make parameter estimates noisy. To accommodate for that, we use data from a large number of participants. Because our hypotheses were tested using hierarchical models, variance in parameter estimates at the first level can be compensated for with additional participants, when testing hypotheses at the second level.

For each model, parameters were estimated and optimized for each participant separately using the *fminsearch* function in MATLAB by minimizing the difference between the predicted and measured SCR to the CS (*Michalska et al., 2017*). We used a series of initial values for the learning rate between 0.1 and 0.8. There were no parameter constraints for the acquisition data fits, as estimates fell within a reasonable range of –1 to 1. We also carried out parameter recovery and found that while some parameters were reasonably well recovered, for the more complex models, many were not. We therefore conducted model comparison for the models with recoverable parameters. We defined a recoverable model as one for which all the parameters had a correlation coefficient of at least 0.20. The limited ability to recover parameters is likely due to the limited number of trials available to estimate the parameters. See Appendix 1 for a full description of model and parameter recovery.

For all models, we calculated the Bayesian Information Criterion (BIC) (*Schwarz, 1978*), which offers a trade-off between model fit and model complexity, for each model. BIC was calculated under a Gaussian assumption as:

$$BIC = -kln(n) + nln(variance) \tag{9}$$

where *k* is the number of parameters in the model, *n* is the number of data points in the dataset that was fit (which corresponds to the number of trials in CS+ and CS- data for the given phase being fit), under the assumption that model errors are i.i.d. and normally distributed (*Priestley, 1981*).

We then compared the distributions of BIC values generated for each model across all subjects using *t*-tests. First, the BIC value for each model for each participant was computed. The differences

between sets of BIC values (by model, for all participants) were used to run *t*-tests to determine whether there were statistically significant differences in the distributions of values for each of the models. We also examined the best model for each participant. In this case the BIC values were compared by comparing the fraction of participants best fit by each of the models. Parameters from the best fitting model were then used in analyses of anxiety symptoms and brain structure.

## Extinction phase

We modeled threat extinction rates, whereby responses to the CS+ and CS- cues decrease over trials. No UCS was delivered during the extinction phase. We fit twelve models to the extinction data (the uncertainty models, models 13 and 14, were excluded because no UCS was ever delivered for these trials, and therefore the uncertainty model would never update), using the same procedure as above. Fitting these models to the extinction data with unconstrained learning rates led to poor performance and the learning rates for some of the participants went to extreme ranges to accommodate the data. Therefore, we refit the models with learning rates constrained to an interval of −1: + 1, where model fits would be interpretable. With this approach ~14% of participants had learning rates pinned at −1 and 1, and for ten participants, the models still did not converge to a solution. Although we would not expect a negative learning rate, we allowed for flexibility in learning by allowing the model to converge with a small or negative learning rate if necessary for the model to converge and provide a reasonable solution. Thus, rates were constrained to the range −1: +1. We also conducted parameter recovery for the twelve models used for extinction (see Appendix). The models that had recoverable parameters were included in model comparison using BIC values and the same methods used for the acquisition data.

## Brain imaging

Analyses tested associations among threat learning parameters and imaging measures. MRI images (1 mm$^3$) acquired on a 3-Tesla MR750 GE scanner (32-channel head coil; sagittal; 176 slices; 256 × 256 matrix; 1mm$^3$ isotropic voxels; flip angle = 7°; repetition time (TR) = 7.7 ms, echo time (TE) = 3.42 ms), were collected from 148 of the 215 participants (69%; 81 females, *M* age = 18.38 years) within 90 days of the task. MRI data for a larger sample containing these participants appear in previous reports using different analyses (*Abend et al., 2020*; *Gold et al., 2017*; *Gold et al., 2016*). FreeSurfer (version 6.0, http://surfer.nmr.mgh.harvard.edu) was used for processing, as reported elsewhere (*Abend et al., 2020*). Statistical tests were performed using PALM (Permutation Analysis of Linear Models) (*Winkler et al., 2014*). Surface-based analyses considered whole-brain cortical thickness (10,242 vertices) using the TFCE (threshold-free cluster enhancement) statistic (*Smith and Nichols, 2009*). Analyses of subcortical volumes generated by FreeSurfer (bilateral amygdala, hippocampus, thalamus, diencephalon, caudate, putamen, pallidum, and nucleus accumbens; brainstem) considered gray matter volume (GMV). For each morphometry measure, analyses included global whole-brain estimates of the measure (global average thickness, total intracranial volume) as nuisance, as recommended in prior research (*Nordenskjöld et al., 2015*). Sex was also used as a nuisance variable given reported sex differences in brain structure (*Beesdo et al., 2009*; *Abend et al., 2020*; *Ruigrok, 2014*). Subcortical results were visualized using Blender version 2.90 (*Kent, 2015*).

## Data analysis

Analyses tested associations among parameters of the best-fitting models for threat conditioning and extinction processes, anxiety severity, and brain morphometry measures, as moderated by age. Specifically, for each phase, we first identified the model best accounting for physiological responses. For each estimated parameter in this winning model, we next examined whether it was associated with anxiety severity, and the moderation of this association by age, using a single regression model. Conditioning effects were best modeled using four parameters (see Results), and thus significance level for each tested effect was determined via Bonferroni at α = 0.05/(4 parameters) = 0.0125. Extinction effects were best modeled using one parameter (see Results), and thus significance level for each tested effect was determined at α = 0.05/(2 parameters) = 0.025. Next, we examined whether brain structure moderated the associations between each learning parameter and anxiety severity. As noted above, analyses considered whole-brain cortical thickness (at the vertex level) and subcortical GMV (at the structure level), and used FWE rate correction for multiple comparisons of α < 0.05, whereby the

family of tests for each analysis consisted of all vertices/subcortical structures across all tested effects, as above.

Sample size was not determined by a power analysis, for two reasons. First, no prior data exist on modeling of the processes of interest and their associations with anxiety severity; second, we aimed, a priori, to recruit a sample that was substantially larger than prior studies on threat learning in anxiety (*Duits et al., 2015*; *Dvir et al., 2019*).

## Results
### Raw SCR data

Raw SCR data by trial and phase are depicted in *Figure 1*. Repeated-measures ANOVA of SCR data with Phase (conditioning, extinction) and CS (CS-, CS+) as within-subject factors and Anxiety severity (Z-scores) and Age (years) as between-subject factors, indicated a significant Phase×CS interaction, $F(1,212)=10.97$, $p = 0.001$. Follow-up analyses indicated greater response to CS+ than CS- during conditioning, $F(1,212)=24.24$, $p < 0.001$, but not during extinction, $F(1,212)=0.24$, $p = 0.631$, demonstrating successful conditioning and extinction in the task. Anxiety and age did not significantly moderate the Phase×CS interaction, $ps > 0.05$.

Additional lower-order effects were noted. We noted a main effect of Phase, $F(1,212)=10.29$, $p = 0.002$, with greater response during conditioning than during extinction. Further, we noted a main effect of CS, $F(1,212)=11.93$, $p < 0.001$, with greater response to CS+ than to CS- across the task. These effects were qualified by a significant Phase×Anxiety interaction, $F(1,212)=6.25$, $p = 0.013$; follow-up analyses indicated no main effect of Anxiety during conditioning, $F(1,212)=0.14$, $p = 0.712$, and a trend-level main effect of Anxiety during extinction, $F(1,212)=3.30$, $p = 0.074$. Finally, we noted a main effect of Age, $F(1,212)=25.42$, $p < 0.001$, with decreasing response with greater age.

### Reinforcement models
#### Conditioning phase

We fit a series of reinforcement learning models to the conditioning data (see *Figure 2—figure supplement 1*). Our goal was to provide accurate fits of the SCRs, along with parameter estimates that could be related to symptom and structural brain data. In all cases, we used the response to the UCS, on each trial in which it occurred, to generate prediction errors, δ(t), that were used to update CS+/- predictions (see below). We built a family of models that included different features and their combinations, as well as a hybrid Rescorla-Wagner Pearce-Hall models (*Li et al., 2011*) and uncertainty models with and without habituation (*Tzovara et al., 2018*). We characterized these models and our model fitting process by carrying out parameter and model recovery (see Appendix 1). The models for which parameters could not be recovered were removed from model comparison. After this step, six models remained (models 1, 4, 5, 7, 8, 13, 14).

To examine the quality of the fits of the remaining models, we calculated BIC values for each model for each participant, and based on the BIC values, determined which model was best for each participant (*Figure 2A*). As these models are related, and we had a broad sample of different responses to the conditioning task, there was not a dominant model. However, we used the BIC values to define a best model at the population level.

To characterize model fits at a population level, we first performed a repeated-measures ANOVA on BIC values (model as fixed effect, participant as random effect, BIC value as dependent variable). We found that BICs varied across models, $F(1,5)=22.54$, $p<0.001$. We further found that models 7 and 8 performed better than all other models based on post-hoc comparisons ($ps<0.05$, Bonferroni-corrected). These models did not, however, statistically differ ($p=0.20$). Since model 7 is a simpler model, we chose it as the optimal model for our data and used its four free parameters (for CS+, α: threat learning rate, φ: habituation rate; for CS-, α: threat generalization rate, φ: safety learning rate) in subsequent analyses; see *Figure 2*. See Appendix 1 for model fit when the sample is split into subgroups based on youth vs adult and patient vs healthy control participants.

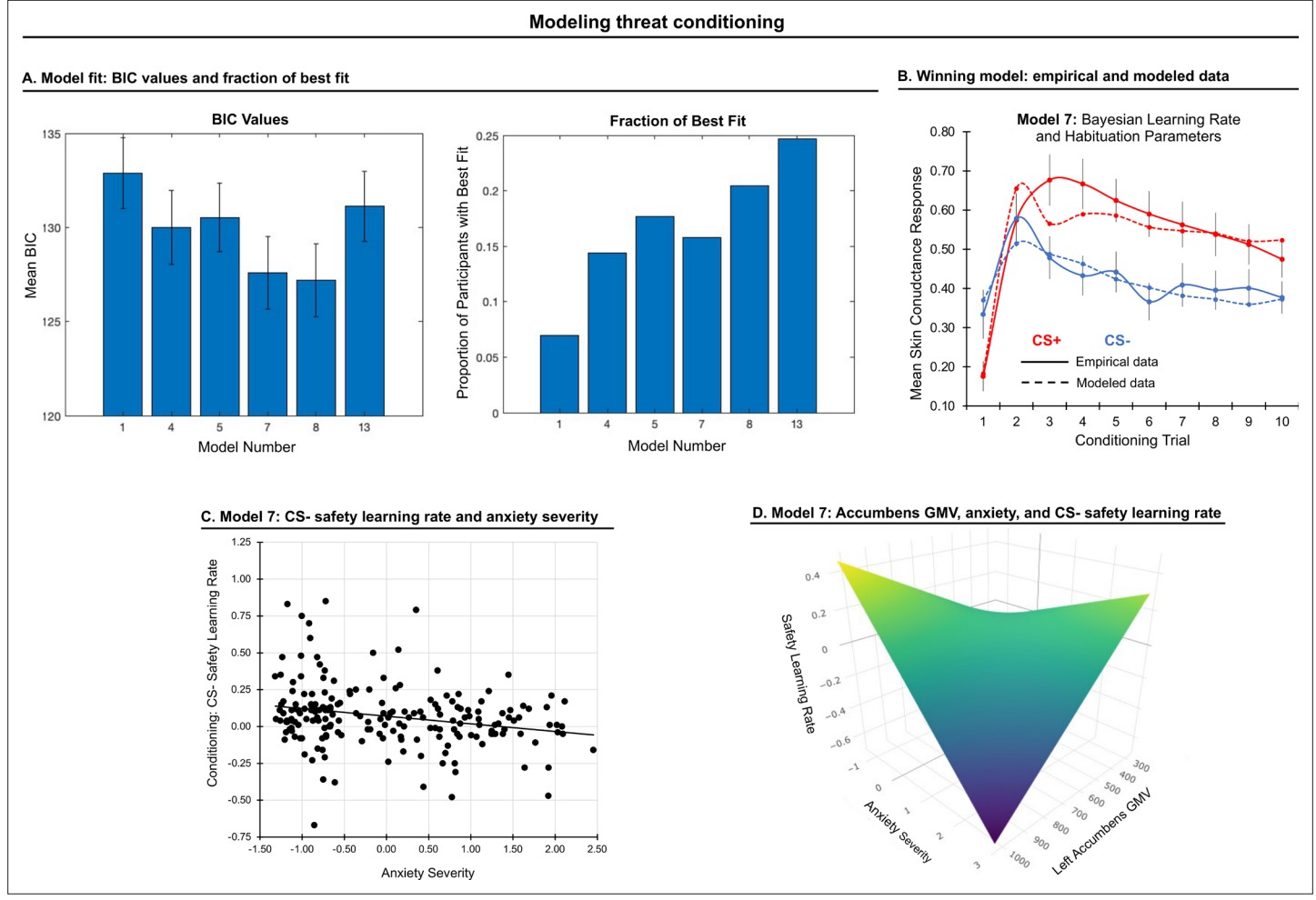

**Figure 2.** Modeling threat conditioning. (**A**) Bars in left panel depict BIC values for each of the fourteen models fit to the CS- and CS + conditioning data. Error bars indicate one standard error of the mean. Bars in right panel depict the proportion of participants for whom each model provided the best fit. (**B**) Based on model fit indices, model 7 was chosen as the best-fitting model for conditioning data. Graphs depict empirical skin conductance data (full line) and fitted data (dashed line) for model 7 fitted to CS+ (red) and CS- (blue) conditioning data. Data are smoothed for display purposes only. (**C**) Association between model 7's CS- learning rate parameter and anxiety severity. (**D**) The association between model 7's CS- learning rate parameter and anxiety severity was moderated by left accumbens gray matter volume (GMV).

The online version of this article includes the following figure supplement(s) for figure 2:

**Figure supplement 1.** Threat conditioning model fits.

**Figure supplement 2.** Conditioning model recovery.

**Figure supplement 3.** Proportions of participants for whom each model provided the best fit, when the sample is divided into anxiety (patients vs healthy comparisons) and age (youth vs adults) groups.

## Conditioning: CS+ threat learning and habituation

### Anxiety

We examined associations between anxiety severity and CS+ threat learning rate and habituation rate (generated by model 7). These rates did not significantly correlate with anxiety severity, ßs < 0.09, *p*s > 0.193.

### Brain structure

No significant associations emerged between threat learning or habituation rate and brain structure, all $p_{FWE}$s > 0.05.

## Conditioning: CS- threat generalization and safety learning

### Anxiety

We examined associations between anxiety severity and CS- parameters generated by model 7: generalization of fear response to the non-reinforced CS- (threat generalization) and reduction in response to it through non-reinforcement (safety learning). Anxiety severity was negatively associated with safety learning rate, indicating slower safety learning with greater anxiety severity, $\beta = -0.243$, $p = 0.001$; see *Figure 2C*. This effect remained significant when controlling for CS+ habituation rate, $\beta = -0.239$, $p = 0.001$, indicating the specificity of the association between anxiety and safety learning (as opposed to a general habituation effect). No other effects were observed.

### Brain structure

A significant interaction between anxiety severity and left accumbens GMV on safety learning rate emerged, $p_{FWE} = 0.043$. This effect remained significant when controlling for CS+ habituation rate, $p_{FWE} = 0.040$. Decomposition of this interaction indicated that when accumbens volume was smaller, slower safety learning was associated with greater anxiety severity; see *Figure 2D*.

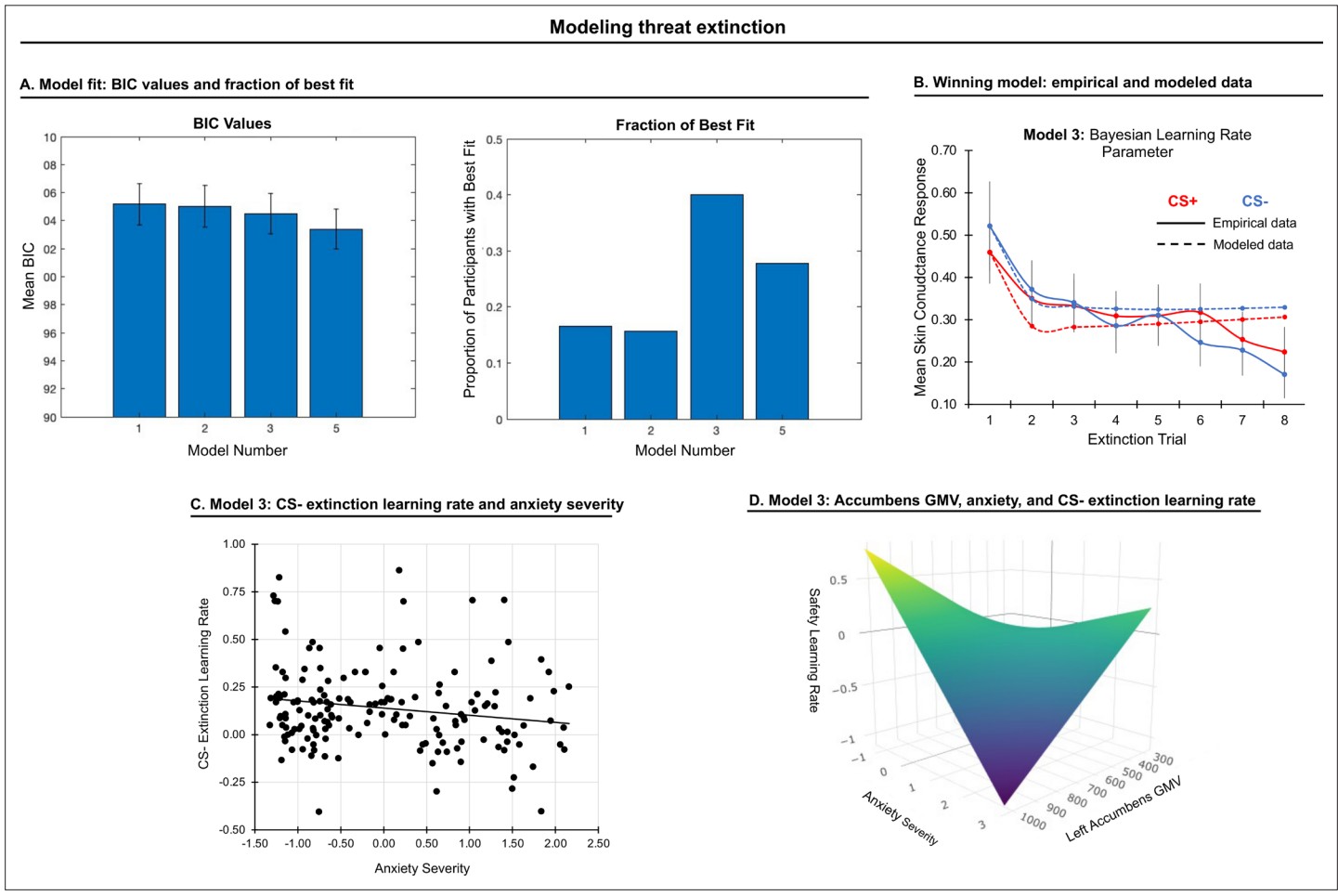

**Figure 3.** Modeling threat extinction. (**A**) Bars in left panel depict BIC values for each of the twelve models fit to the CS- and CS+ extinction data. Error bars indicate one standard error of the mean. Bars in right panel depict the proportion of participants for whom each model provided the best fit. (**B**) Based on model fit indices, model 3 was chosen as the best-fitting model for extinction data. Graphs depict empirical skin conductance data (full line) and fitted data (dashed line) for model 3 fitted to CS+ (red) and CS- (blue) extinction data. Data are smoothed for display purposes only. (**C**) Association between model 3's CS- extinction rate parameter and anxiety severity; this association is only trend-level significant. (**D**) The association between model 3's CS- learning rate parameter and anxiety severity was moderated by left accumbens gray matter volume (GMV); this association is only trend-level significant.

The online version of this article includes the following figure supplement(s) for figure 3:

**Figure supplement 1.** Threat extinction model fit.

### Extinction phase

#### Model fit

Extinction data from each participant were fit with the first twelve models used to model acquisition responses (See *Figure 3—figure supplement 1*). Four models with parameters that could be recovered were included in model comparison (models 1, 2, 3, and 5). A repeated-measures ANOVA on BIC values (model as fixed effect, participant as random effect), yielded a significant effect of model, $F(1,3)=15.45$, $p < 0.001$. Model 3 was picked most often as the best-fitting model (*Figure 3*), and therefore we compared it with other models providing better fit using *t*-tests. However, model 3 did not differ statistically from models 1, 2, or 5 (*ps* > 0.05, Bonferroni-corrected). Given its relative simplicity and the fact that it was frequently chosen across participants, we used the parameters from model 3 (Bayesian-decay learning rate) to represent the extinction process, with one learning rate for CS+ and one for CS-.

#### Anxiety

CS+ extinction rates did not correlate with anxiety severity. CS- extinction rate marginally correlated with anxiety severity, $\beta = -0.16$, $p = 0.034$, such that slower extinction was associated with greater anxiety; see *Figure 3*. A comparable effect was noted when controlling for CS+ extinction rate, $p = 0.031$, as well as when controlling for CS- learning and habituation rates during conditioning, $p = 0.030$. No other effects were observed.

#### Brain structure

Gray matter volume in left accumbens moderated the association between CS- extinction rate and anxiety severity at trend level, $p_{FWE} = 0.062$, such that as accumbens volume was smaller, slower extinction was associated with greater anxiety severity; see *Figure 3*. No other effects were observed.

In addition to the primary analyses examining associations between anxiety severity and learning parameters and the moderation of these associations by brain structure across the full sample, we also examined these effects separately among youth and adult participants. These are reported in full in Appendix 1. Briefly, they suggest that observed anxiety effects on safety learning and extinction are attenuated with age. Further, these suggest moderation of anxiety-safety learning rate by several structures (bilateral nucleus accumbens, brain stem, and amygdala) in youth participants but not in adults. In contrast, extinction rates were associated with amygdala, accumbens, and brainstem volume moderation in adults, but not in youths; see Appendix 1.

## Discussion

This study modeled trial-by-trial psychophysiological data to quantify threat learning processes and their associations with anxiety symptoms and neuroanatomy (*Mineka and Oehlberg, 2008*; *Duits et al., 2015*; *Fanselow, 2018*; *Casey et al., 2015*). Several findings emerged. First, threat conditioning was optimally modeled with a RW model that featured Bayesian learning rate decay and a habituation rate. Within the conditioning process, slower safety learning rate was found to relate to more severe anxiety, and nucleus accumbens volume moderated this association. Finally, extinction learning was best modeled using a Bayesian (diminishing) learning rate decay and extinction of response to the safety, but not threat, stimulus was associated with greater anxiety. This study identifies specific latent threat-learning parameters that relate to anxiety symptom severity and neuroanatomical features.

The optimal fit for threat conditioning was a RW model that assumes Bayesian learning decay and habituation terms. A RW learning rule has long been proposed to underlie reinforcement learning, whereby point estimates of future outcomes are calculated by modifying current predictions according to prediction error, as weighted by a fixed learning rate (*Rescorla and Wagner, 1972*). Models based on a RW rule have been shown to describe the acquisition of conditioned fear responses in animals and humans (*Herry and Johansen, 2014*). Here, we show that an incrementally-diminishing threat-learning rate provides a better fit to physiology data (i.e., SCR) in humans, suggesting that conditioning of threat-anticipatory physiological responses is governed by a set learning parameter that is weighted less in proportion to additional reinforcement trials. It therefore places greater emphasis on learning from initial encounters with danger, supporting an adaptive role for this learning process in environments that contain threats (*Fanselow, 2018*).

The inclusion of a habituation term could indicate that conditioned responding decreases as the threat cue becomes more predictable, beyond what is explained by the basic RW model itself (*Rescorla and Wagner, 1972*). This could be a product of the 80% reinforcement schedule used here; a different reinforcement ratio could have affected this habituation term. Additionally, the habituation term could reflect our use of trial-level UCS value when modeling, rather than 0/1 as is often done, such that the magnitude of response to the UCS contributes to learning, rather than just its categorical absence/presence. Indeed, response to UCS in conditioning paradigms is often shown to naturally diminish as learning is achieved, potentially reflecting effective and reliable anticipation of it (*Britton et al., 2013*; *Goodman et al., 2018*; *Johansen et al., 2010*). Further, in other work (*Abend, 2021*) we show that increasing UCS potency results in increasing physiological response to the UCS. The response habituation term in the model may therefore account for individual differences in dynamic changes in the strength of response that both CS+ and UCS elicit.

This study extends prior work by modeling multiple learning effects that were evident in the data during threat conditioning and showed associations with anxiety severity and neuroanatomy. The introduction of an aversive UCS led to generalization of conditioned fear response to the CS- occurring most strongly early in conditioning, and followed by safety learning as this generalized response was inhibited by the CS- not predicting the aversive outcome. Learning theories of anxiety posit that each of these processes might play a role in the emergence and maintenance of pathological anxiety (*Mineka and Oehlberg, 2008*; *Duits et al., 2015*; *Dymond et al., 2015*; *Vervliet et al., 2013*; *Lissek et al., 2014*). Our modeling approach allowed us to test these theories. First, threat generalization was noted in the data, indicating that this process can be usefully modeled with this approach. However, the rate of generalization was not significantly associated with anxiety severity. Second, our findings provide support for propositions (*Mineka and Oehlberg, 2008*; *Lissek et al., 2009*) that link greater anxiety with slower safety learning rates (*Mineka and Oehlberg, 2008*; *Craske et al., 2018*). This association remained significant when controlling for CS+ habituation during conditioning, indicating that it was not a general habituation process. This finding suggests that fear responses to aversive events may initially generalize to neutral proximal stimuli, as a potentially adaptive mechanism, but that continued responding to these stimuli may contribute to the persistent and generalized fears associated with anxiety (*Lissek et al., 2014*). Importantly, the use of computational modeling enabled us to extend prior work by directly examining associations between learning *rates* and anxiety.

Our data suggest that the association between anxiety and safety learning was moderated by nucleus accumbens volume. Recent work indicates a role for this structure in regulating fear responses and threat-safety discrimination, including in the context of extinction learning (*Ray et al., 2020*; *Dutta et al., 2021*; *Abraham et al., 2014*). Our findings extend such work by suggesting that variation in accumbens structure relates to fear regulation following fear generalization to safety cues. Importantly, this effect is associated with anxiety severity, indicating potential pathophysiological relevance. These findings encourage more research on accumbens structure in predicting future anxiety as well as functional imaging work linking its structure and function in the context of safety learning.

Our results show that threat extinction learning also follows an incrementally-diminishing learning rate. As in threat generalization and safety learning, this is the first study to directly quantify the temporal dynamics of this learning process via modeling. We show an association between slower extinction of the safety stimulus and greater anxiety severity, providing some support for major theories of anxiety and its treatment that highlight extinction processes (*Mineka and Oehlberg, 2008*; *Duits et al., 2015*; *Milad and Quirk, 2012*; *Barlow, 2002*; *Craske et al., 2018*; *Pittig et al., 2016*; *Vervliet et al., 2013*; *Papalini et al., 2020*). Thus, in the context of exposure to conditioned stimuli during threat extinction, individuals with higher relative to lower anxiety symptoms diminish fear responses to safe stimuli more slowly. Importantly, this association was maintained when controlling for threat extinction rate as well as safety learning rate during conditioning, indicative of specificity. As in safety learning, accumbens structure moderated anxiety-learning associations. Thus, associations with anxiety severity, and moderation by accumbens anatomy, emerged for two processes in which the safe value of the CS- is to be learned, highlighting the role of response regulation to such stimuli in anxiety symptoms. It should be emphasized, however, that effects for extinction were weaker than those observed during conditioning and were evident only at trend-level, calling for caution in its interpretation and for replication.

Of note, prior work identified the RW-PH "hybrid" model as providing optimal fit to data for the expression of threat contingencies among healthy adults as well as adults with PTSD (*Homan et al., 2019*; *Tzovara et al., 2018*; *Zhang et al., 2016*). Here, we found that this model did not provide the best fit for rapid threat conditioning or extinction among the models tested. This discrepancy could potentially be attributed to the nature of threat learning processes studied. While we focused on rapid acquisition/extinction of threat contingencies which takes place over a short training schedule (*LeDoux, 2000*; *Fanselow, 2018*), prior modeling studies examined threat contingencies over much longer durations (over 80 CS+ trials). The hybrid model includes a cumulative associability term which could be more sensitive to tracking contingency values or the expression of conditioned fear as it is optimized over longer durations. Thus, differences in rapid, crude acquisition vs optimized expression of threat contingencies processes could account for model differences. As such, the current and prior work could be seen as complementary in terms of threat learning processes studied, and both sets of findings could usefully inform study design for future work.

While the design of this study offers a unique opportunity to examine age effects on threat learning as these relate to anxiety severity, an inherent challenge that arises in research on anxiety along the lifespan is how to combine anxiety data from youth and adult participants. Although the SCARED and STAI are each considered 'gold standard' measures in their respective target populations, they nevertheless are not identical. Under the assumption that these capture similar constructs, we uncovered several anxiety effects across the full age range. Alternatively, one may consider these measures to be incomparable, in which case analyses should be restricted to specific age groups. This alternative approach indicated an attenuation of anxiety-learning associations with age, as well as age group-specific moderation of these associations by brain structure. Given that age-dependent effects appear to emerge more strongly when age groups were considered separately, future research could emphasize this approach. When hypotheses call for analyses across the lifespan, consideration of optimal harmonization of clinical data is needed.

Our prior report used this dataset to examine links among associative threat learning, anxiety, and neuroanatomy using a 'standard' analytical approach whereby responses to CSs are averaged across trials (*Abend et al., 2020*). Here, we utilized a reinforcement modeling framework to test such links; this approach importantly extends our prior work in several ways. First, the approach applied here enabled us to *directly* quantify the temporal dynamics of CS and UCS associations that are at the core of associative learning (as opposed to measuring only responses to CSs and treating all trials as equivalent), thereby providing a more sensitive measure of threat learning processes, and how these relate to anxiety. Second, the current approach allowed us to quantify *multiple characteristics* evident in the learning processes (e.g. acquisition and habituation rates). As such, the modeling approach may be more sensitive to detect different, specific aspects of learning; this, in turn, enabled us to simultaneously test different theories linking these specific threat learning aspects and anxiety (e.g., slower safety learning) with greater sensitivity. Indeed, whereas our previous report failed to identify such associations, the current report provides novel insight on the pathophysiology of anxiety by linking variations in distinct threat learning processes to anxiety symptoms and subcortical structures. This distinction therefore highlights the utility of using a reinforcement learning framework to test questions on threat learning and anxiety.

Along these lines, the findings generated from this application of a computational approach to link biological, clinical, and imaging data could initiate continued research along several lines. In terms of clinical research, the identified associations between threat learning processes and anxiety severity could inform theories on the etiology of anxiety symptoms and guide studies that aim to further qualify the learning conditions and mechanisms that promote anxiety (*Li and McNally, 2014*; *Vervliet et al., 2013*). Further, influential theories that place threat extinction processes at the center of exposure-based therapy (*Milad and Quirk, 2012*; *Craske et al., 2018*; *Pittig et al., 2016*; *Papalini et al., 2020*) could be evaluated more sensitively using modeling-derived indices, while such treatment approaches could incorporate insight on safety learning (*Craske et al., 2018*). Translational research on mechanisms of threat learning could also benefit from these findings. Thus, neuroscience research in humans could focus on the specific processes identified here and extend our structural imaging findings to insight on function within this circuitry. Research in animals could complement such work by further delineating, via invasive manipulations (e.g., *Likhtik and Paz, 2015*), the roles in threat learning of the subcortical structures identified here.

Several important limitations should be acknowledged. First, modeling was based on relatively few task trials; while we focused on the specific processes of rapid learning of threat associations that take place over very few pairings (*Fanselow, 2018*), this could have led to noisier parameter estimates that could diminish accuracy and statistical power. Second, this was a cross-sectional study; a longitudinal design examining whether learning rates predict later emergence of symptoms would allow stronger inferences about developmental processes (*Lonsdorf and Merz, 2017*). Third, only structural MRI data were examined; functional imaging during threat learning could reveal additional correlates of learning circuitry (*Homan et al., 2019*), although the delivery of aversive stimuli to participants with anxiety in the MRI scanner presents a challenge (*Thorpe et al., 2008*). Fourth, SCR data were analyzed using a single method, which, while established, relies only on directly-observable effects; future studies may consider using novel, computational analysis methods which could potentially reveal effects not observed using the current method (*Bach et al., 2018*; *Bach and Friston, 2013*; *Bach et al., 2020*; *Ojala and Bach, 2020*). Along these lines, a multiverse approach may be used in future work to comprehensively compare multiple methods of quantifying threat learning. Fifth, while we used strict exclusion criteria to reduce effects of psychopathology other than anxiety, this does not eliminate all such potential effects (particularly when using the STAI *Knowles and Olatunji, 2020*); future research may wish to utilize computational approaches (e.g. bifactor models) to estimate symptom variability that is unique to anxiety (*Tseng et al., 2021*). Sixth, future research may consider alternative psychophysiology measures to SCR which may potentially be more sensitive to threat learning effects (*Ojala and Bach, 2020*). Finally, there were differences in sex and age between individuals included and excluded (non-responders) from analyses, as reported previously (*Boucsein et al., 2012*; *Bari et al., 2020*); while these differences did not influence our findings, they could still limit generalizability and thus future studies should consider such potential differences. Several strengths partially mitigate these limitations and address general shortcomings in threat learning research (*Lonsdorf et al., 2017*; *Ney et al., 2018*). First, the large sample size increases precision of estimated parameters (*Asendorpf et al., 2020*), offsetting, to some extent, the small number of trials used for modeling. Second, participants were carefully assessed and free of medications known to impact threat learning and psychophysiology (*Lonsdorf et al., 2017*). Third, wide anxiety-symptom and age ranges generate inferences with reasonable statistical power. Fourth, task and setting were identical for all participants, reducing measurement confounds and noise.

As a final technical comment, reinforcement learning model parameters were estimated using maximum likelihood techniques on individual subjects followed by model comparison. Future work could expand on this by using hierarchical Bayesian parameter estimation to reduce the variance around parameter estimates (*Piray et al., 2019*; *van Geen and Gerraty, 2021*; *Lee and Newell, 2011*). However, choosing prior distributions within the hierarchical Bayesian approach is not trivial and may not work for all of the models tested in this study. As such, future work could focus on fewer models, such as those that survived parameter recovery in this study, test a range of priors, and determine whether parameter estimation could be improved.

In conclusion, in this study we used computational modeling to index dynamic learning processes associated with threat learning. Through this modeling approach, we quantified these learning processes, revealed specific associations with anxiety symptoms, and identified neuroanatomical substrates. These findings extend our knowledge of how these learning processes manifest in humans and how variations in these could potentially contribute to anxiety symptomatology.

## Acknowledgements

We thank the participants and families, as well as the staff of the Intramural Research Program of the National Institute of Mental Health (IRP, NIMH), National Institutes of Health. We also thank Emily Ronkin, Elizabeth Steuber, Madeline Farber, Jessica Sachs, Brigid Behrens, Carolyn Spiro, and Omri Lily for their contribution to data collection.

This research was supported (in part) by the NIMH IRP (ZIAMH002781-15; DSP), NIH grant K99/R00MH091183 (JCB), and a NARSAD Young Investigator Grant from the Brain & Behavior Research Foundation (RA).

## Additional information

### Funding

| Funder | Grant reference number | Author |
|---|---|---|
| National Institutes of Health | ZIAMH002781-15 | Daniel S Pine |
| National Institutes of Health | R00MH091183 | Jennifer C Britton |
| Brain and Behavior Research Foundation | 28239 | Rany Abend |

The funders had no role in study design, data collection and interpretation, or the decision to submit the work for publication.

### Author contributions

Rany Abend, Conceptualization, Formal analysis, Investigation, Methodology, Project administration, Software, Supervision, Validation, Visualization, Writing - original draft, Writing – review and editing; Diana Burk, Formal analysis, Methodology, Software, Visualization, Writing – review and editing; Sonia G Ruiz, Data curation, Formal analysis, Methodology, Project administration, Visualization, Writing - original draft, Writing – review and editing; Andrea L Gold, Conceptualization, Data curation, Formal analysis, Investigation, Methodology, Supervision, Writing - original draft, Writing – review and editing; Julia L Napoli, Anderson M Winkler, Formal analysis, Investigation, Methodology, Software, Visualization, Writing – review and editing; Jennifer C Britton, Conceptualization, Data curation, Investigation, Methodology, Project administration, Supervision, Writing - original draft, Writing – review and editing; Kalina J Michalska, Data curation, Investigation, Methodology, Project administration, Supervision, Writing - original draft, Writing – review and editing; Tomer Shechner, Data curation, Investigation, Project administration, Supervision, Writing – review and editing; Ellen Leibenluft, Methodology, Supervision, Writing – review and editing; Daniel S Pine, Conceptualization, Funding acquisition, Investigation, Project administration, Resources, Supervision, Writing – review and editing; Bruno B Averbeck, Conceptualization, Formal analysis, Investigation, Methodology, Project administration, Software, Supervision, Writing - original draft, Writing – review and editing

### Author ORCIDs

Rany Abend (ID) http://orcid.org/0000-0003-0022-3418
Diana Burk (ID) http://orcid.org/0000-0001-7775-2989
Bruno B Averbeck (ID) http://orcid.org/0000-0002-3976-8565

### Ethics

Written informed consent was obtained from adult (age greater than or equal to 18 years) participants as well as parents, and written assent was obtained from youth (age under 18). Procedures were approved by the NIMH Institutional Review Board (protocol 01-M-0192).

### Decision letter and Author response

Decision letter https://doi.org/10.7554/eLife.66169.sa1
Author response https://doi.org/10.7554/eLife.66169.sa2

## Additional files

### Supplementary files

- Transparent reporting form
- Source code 1. Code for reinforcement learning models.
- Source code 2. Code for imaging processing and analysis.

### Data availability

We cannot share the full dataset due to the NIH IRB requirements, which require participants to explicitly consent to their data being shared publicly. An important element in that is to protect patients who

agree to participate in studies that relate to their psychopathology. Such consent was not acquired from most participants; as such, we cannot upload our complete dataset in its raw or deidentified form, or derivatives of the data, since we will be violating IRB protocols. Still, a subset of participants did consent to data sharing and we have uploaded their data as noted in the revised manuscript (https://github.com/rany-abend/threat_learning_eLife, copy archived at swh:1:rev:221919cc02c4b5ae00f-c21764095b8b4d2a14201). Researchers interested in potentially acquiring access to the data could contact Dr. Daniel Pine (https://pined@mail.nih.gov/), Chief of the Emotion and Development Branch at NIH, with a research proposal; as per IRB rules, the IRB may approve adding such researchers as Associate Investigators if a formal collaboration is initiated. No commercial use of the data is alowed. The modeling and imaging analyses have now been uploaded in full as source code files.

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

## Appendix 1

## Methods

### Participants

Prior to participation in the study, patients' psychiatric symptoms were assessed on three separate occasions, via (1) telephone screen with a psychiatric nurse, (2) in-person, standardized diagnostic assessment (see below) with a trained clinician, and (3) independent assessment and confirmation of diagnosis by a senior psychiatrist. All patients agreed to enter treatment for their anxiety disorder; as such, the patient data reported reflect populations of youth and adults with anxiety disorders.

Individuals were included if they were medication-free, physically healthy, and had an IQ>70, based on the Vocabulary and Matrix Reasoning subscales of the Wechsler Abbreviated Scale of Intelligence (*Wechsler, 1999*). All pediatric patients in the anxiety group had to suffer from generalized anxiety, social anxiety, and/or separation anxiety disorders as their major source of psychiatric impairment and need for treatment. Adult patients were eligible for any of the same ongoing three anxiety disorders as well as panic disorder and were not required to be seeking treatment. A diagnosis of major depressive disorder, bipolar disorder, obsessive compulsive disorder, disruptive mood dysregulation disorder, or posttraumatic stress disorder was exclusionary. Patients with anxiety were permitted to have comorbid additional anxiety disorders or attention-deficit/hyperactivity disorder if presenting as a secondary, minor problem, relative to the primary diagnosis. Healthy participants were diagnosis-free. Exclusion criteria for both groups included current psychotropic medications, inclusion of family relatives in the study, or physical health problems.

Data from 32 participants were excluded due to aborting the task (22 anxious, 8 healthy) or technical problems (1 anxious, 1 healthy). Data from 4 additional participants (2 anxious, 2 healthy) were excluded from analyses since they inquired and were then informed of the CS contingencies prior to the conditioning phase (*Mechias et al., 2010*).

### Diagnosis

Diagnosis of an anxiety disorder was given by trained clinicians using the Kiddie Schedule for Affective Disorders and Schizophrenia for School-Age Children-Present and Lifetime Version (KSADS-PL) (*Kaufman et al., 1997*) for youths (n=140; age <18 years) and the Structured Clinical Interview for DSM-IV-TR Axis I Disorders (SCID) (*First et al., 2002*) for adults (n=75; age ³18 years). All clinicians were trained on an initial series of recorded interviews and then were regularly monitored through a review of interview tapes and reassessments of patients.

### Anxiety severity scores

As noted in the main text, we used dimensional anxiety severity scores in analyses derived from Z-scores on standardized anxiety questionnaires. As can be seen in *Figure 1—figure supplement 1*, severity scores were unimodal and continuous across the sample. Supplemental analyses considered youth and adult participants separately.

### Threat conditioning and extinction task

A schematic representation of the threat conditioning and extinction task is provided in *Figure 1*. The task consisted of a pre-conditioning phase, a conditioning phase, and an extinction phase. Each conditioned stimulus (CS+ and CS-) was presented for 7s. During the pre-conditioning phase, each CS was presented four times to allow physiological responses to the novel stimuli to habituate. During the threat conditioning phase, each CS was presented 10 times, and the CS+ was followed by the UCS with an 80% reinforcement schedule. Participants were instructed that they could learn to predict when the UCS would occur, but they were not explicitly informed of this contingency. During the extinction phase, the CSs were each presented eight times in the absence of the UCS. Throughout all phases, presentation order of the CSs and an inter-trial interval (a gray screen presented for 8-21s, averaging 15s) was pseudo-randomized (two different orders counterbalanced across participants). The task was programmed and administered using PsyLab psychophysiological recording system (PsyLab SAM System, Contact Precision Instruments, London).

Following extraction of SCR data as specified in the main text, we cleaned the data at the trial level. This was of particular importance in this study, since models were fit to trial-by-trial data, and outliers would skew model estimates. Missing skin conductance trial values and within-subject outliers were identified per subject, for each CS type during conditioning and extinction phase separately; outliers

were defined and up to three consecutive values were linearly interpolated using the default settings in the *zoo* package in R[66]. Next, trailing missing trial values during conditioning and leading missing values during extinction were linearly extrapolated. Interpolation and extrapolation of missing data was done, as opposed to censoring, since model generation relied on limited trial-by-trial data (per subject), and learning effects were very rapid, and spurious estimates due to censoring data points was a concern. Subjects with more than 50% missing trial values (i.e., skin conductance = 0mS) for CS+ trials during conditioning were excluded. Then, overall CS+ and CS- skin conductance responses were averaged for each subject. Group outliers for each CS type were identified and excluded. Based on these conservative criteria, skin conductance data from 136 participants were excluded from modeling (52 anxious, 84 healthy).

This proportion of exclusion is similar to excluded data in prior work (*Homan et al., 2019*; *Lonsdorf et al., 2019*). See *Appendix 1—table 1* for demographic and clinical differences as a function of inclusion/exclusion. No significant differences in exclusion were noted for anxiety severity (by diagnosis or continuously) or IQ, but excluded relative to included participants had a higher mean age and a greater proportion of females. Note that age was included in tested models, and thus observed anxiety effects are independent of age effects; further, results did not change when using sex as a covariate. Based on our experience, we believe that this exclusion proportion reflects challenges of physiological data collection during uninstructed aversive tasks, particularly across a wide age range and in those diagnosed with anxiety disorders, in conjunction with conservative exclusion rules since modeling required sufficient trial-by-trial data. These considerations should be taken into account in future research, particularly in studies across a wide age range and in those that consider sex, and require sufficient trial-level data. Implementing newer analytic techniques and equipment, as well as use of more robust aversive stimuli, could potentially mitigate this issue.

In addition to skin conductance, startle electromyography (EMG) and electrocardiography were recorded, but not analyzed in the current report. For the startle EMG measure, startle probes (i.e., 40ms, 4-10 psi of compressed air delivered to the forehead) were presented during the CS trials (but not within the SCR response window, 5-6 seconds post-stimulus onset) and during the inter-stimulus interval.

## Reinforcement modeling

When constructing the models, we used the magnitude of response to the UCS as the reinforcement term, and not the mere presence/absence of the UCS (0/1). This is due to two reasons. First, the Rescorla-Wagner model is best for reinforcement learning processes where there is convergence to a level of performance, as the function asymptotes to the expected value of the reinforcement. Thus, this model naturally asymptotes and fits asymptotic choice (performance) behavior well. When used in the current context, the skin conductance data do not asymptote between a fixed range (i.e., 0 to 100% in the case of choice performance) and therefore the scale must be considered. To use 0/1 modeling of UCS delivery, one would need to additionally normalize the skin conductance for each participant or have a unique scaling factor for each participant. Doing this would be inaccurate given the limited amount of data available for each participant and the presence of noise in SCR data in general. Second, it is important to note that there is substantial decrease in response to the UCS over trials, $F(7,2422)=45.33$, $p<0.001$. This has been noted in other work (*Goodman et al., 2018*). Thus, the value of reinforcement naturally diminishes with presentation, and this should be integrated into the model as reinforcement is not fixed but rather changes with time, potentially affecting learning over trials. As such, we used the UCS response as the reinforcement term.

## Imaging data processing and analysis

All participants underwent MRI scanning at the NIMH Functional Magnetic Resonance Imaging Core Facility. Participants completed a high-resolution, T1-weighted magnetization-prepared rapid-conditioning gradient-echo scan (MPRAGE) with the following parameters: sagittal conditioning; 176 slices; 256x256 matrix; 1mm³ isotropic voxels; flip angle = 7°; repetition time (TR) = 7.7ms, echo time (TE) = 3.42ms. Imaging was conducted within 90 days of the task

### Image Processing

Surface-based analysis followed the procedures in (*Fischl and Dale, 2000* and *Dale et al., 1999*). T1-weighted images were corrected for magnetic field inhomogeneities, affine-registered to the Talairach-Tournoux atlas (*Talairach and Tournoux, 1988*), and then skull-stripped. White matter

(WM) voxels were identified based on their locations, their intensities, and the intensities of neighboring voxels, and grouped into a mass of connected voxels using a six-neighbor connectivity scheme. A mesh of triangular faces was the constructed using two triangles per exposed voxel face. The mesh was next smoothed based on local intensity in the original images using trilinear interpolation (*Dale and Sereno, 1993*); a second smoothing iteration was then applied, resulting in a realistic representation of the interface between gray and white matter. The external cortical surface was produced by identifying a point where tissue contrast is maximal, maintaining constraints on smoothness and possibility of self-intersection (*Fischl and Dale, 2000*). These surfaces were then parcellated using an automated process into smaller regions (*Fischl et al., 2004*). We used the regions of the atlas developed by *Desikan et al., 2006*. Cortical thickness served as the cortical measure of interest.

The subcortical volume-based analysis stream is designed to automatically preprocess MRI volumes and label subcortical tissue classes (*Fischl et al., 2004*; *Fischl et al., 2002*). First, images were affine-registered to MNI305 space. Next, initial volumetric labeling was conducted and variation in intensity due to the B1 bias field was corrected. Finally, a high-dimensional nonlinear volumetric alignment to the MNI305 atlas was performed, and structures were labeled. These included brainstem and left and right amygdala, hippocampus, thalamus, caudate, putamen, pallidum, and nucleus accumbens. The permutations tests corrected for the number of structures tested (see below).

Bias-corrected images from FreeSurfer, in the same space as the labelled structures, were segmented into gray matter, white matter, and cerebrospinal fluid using the FAST module of FSL. The outputs of FAST are images in which the value at each voxel corresponds to the proportion of the volume of the voxel that is occupied by each of these tissue classes (*Zhang et al., 2001*).

### Analyses

Analyses testing for associations among brain structure, learning rates, anxiety, and age were conducted using PALM (Permutation Analysis of Linear Models *Winkler et al., 2014*). Analyses were based on 1000 permutations, followed by an approximation to the tail of the permutation distribution of the maximum statistic using a generalized Pareto distribution (*Winkler et al., 2016*). For each morphometry measure, analyses included global whole-brain estimates of the measure as nuisance. Thus, for subcortical GMV, we controlled for total intracranial volume; for cortical thickness, we controlled for global average thickness. For the cortical analysis, only the surface vertices that represent actual cortex were included, masking out sub-callosal region of each hemisphere that is included in the surfaces only to ensure the topology of a sphere. Sex also served as a nuisance variable in light of known differences in brain structure (*Abend et al., 2020*; *Ruigrok, 2014*). All analyses used familywise error (FWE) rate correction for multiple comparisons across all contrasts. Subcortical results were visualized using Blender version 2.90 (*Kent, 2015*).

## Results

### Raw SCR data

In addition to the effects on raw SCR data reported in the main text, we also noted a main effect of Phase, $F(1,212)=10.29$, $p=0.002$, with greater response during conditioning than during extinction. Further, we noted a main effect of CS, $F(1,212)=11.93$, $p<0.001$, with greater response to CS+ than to CS- across the task. These effects were qualified by a significant Phase´Anxiety interaction, $F(1,212)=6.25$, $p=0.013$; follow-up analyses indicated no main effect of anxiety during conditioning, $F(1,212)=0.14$, $p=0.71$, and a trend-level main effect of anxiety during extinction, $F(1,212)=3.30$, $p=0.07$. Finally, we noted a main effect of age, $F(1,212)=25.42$, $p<0.001$, with decreasing response with greater age.

### Reinforcement modeling

*Figure 2—figure supplement 1* depicts the empirical and modeled data across the sample for each of the models fit to the CS- and CS+ conditioning data.

We carried out parameter recovery by using the parameters for each model from the participants to whom it was best fit and used those parameters to generate synthetic data from the model. For each model, we then fit the original model to the synthetic data and compared the corresponding actual and "recovered" parameters using correlation coefficients; see *Appendix 1—table 2*. For acquisition, eight models had all parameter correlation coefficients > 0.2; these models were included in model comparison and subsequent analyses.

Parameter recovery was also conducted for the extinction modeling. The same criteria were applied to include only models with correlation coefficients > 0.20 for all model parameters. See *Appendix 1—table 3*.

**Appendix 1—table 1.** Demographics (sex, age, IQ) and anxiety severity (by diagnosis: anxious/ healthy; by continuous anxiety scores on the Screen for Child Anxiety Related Emotional Disorders or ) for participants who were included or excluded from data analysis due to excessive missing data.

Differences between included and excluded participants were tested using chi-squared or independent-samples t-tests.

|  | Excluded | Included | Test Statistic |
|---|---|---|---|
| N | 136 (94 F) | 215 (116 F) |  |
| % Female | 69.11 | 53.95 | $\chi^2_{(1)}$= 7.35, *P*=.006 |
| N Anxiety diagnosis | 55 | 104 | $\chi^2_{(1)}$= 1.81, *P*=.18 |
| N Healthy | 85 | 111 | $\chi^2_{(1)}$= 3.56, *P*=.06 |
| Mean (SD) age | 23.40 (9.16) | 18.76 (9.39) | $t(302.29) = 4.62$, *P*<.001 |
| Mean (SD) anxiety | −0.09 (0.89) | 0.05 (1.06) | $t(288.66) = 1.27$, *P*=.21 |
| Mean (SD) IQ | 114.68 (13.32) | 113.98 (11.90) | $t(271) = .50$, *P*=.61 |

**Appendix 1—table 2.** Corelations between the actual parameters and recovered parameters for each of the 14 models for acquisition data.

Models in bold met the criteria for model comparison (all corelation values for all parameters >0.20). Note that the last row reflects an additional variant of the Tzovara et al. model with two additional habituation parameters (habituation for CS+ and CS- for 6 total parameters) that was examined for completeness (see Methods).

| Model | Parameter 1, CS+ learning Rate | Parameter 2, CS- learning Rate | Parameter 3, CS+ habituation | Parameter 4, CS- habituation |  |  |  |
|---|---|---|---|---|---|---|---|
| 1 | 0.99 | 0.56 |  |  |  |  |  |
| 2 (inertia) | 0.86 | 0.22 |  |  |  |  |  |
| 3 (Bayesian) | 0.52 | 0.21 |  |  |  |  |  |
| **4** | **0.76** | **0.99** | **>0.99** | **0.99** |  |  |  |
| 5 (inertia +Bayesian) | 0.74 | 0.46 |  |  |  |  |  |
| 6 (inertia) | 0.38 | 0.17 | 0.53 | >0.99 |  |  |  |
| **7 (Bayesian)** | **0.59** | **0.42** | **0.98** | **>0.99** |  |  |  |
| **8 (inertia +Bayesian)** | **0.52** | **0.62** | **0.96** | **>0.99** |  |  |  |
|  | Parameter 1, CS +learning Rate | Parameter 2, CS- learning Rate | Parameter 3, CS +learning rate update | Parameter 4, CS- learning rate update | Parameter 5, regression $\beta_o$ | Parameter 6 regression $\beta_1$ | Parameter 7 regression $\beta_2$ |
| 9 | 0.53 | 0.13 | 0.52 | 0.02 |  |  |  |
| 10 | −0.09 | 0.04 | −0.37 | 0.46 | 0.35 | 0.21 |  |
| 11 | 0.49 | 0.07 | −0.37 | 0.46 | 0.35 | 0.21 |  |
| 12 | 0.34 | 0.04 | −0.39 | 0.66 | 0.002 | 0.02 | 0.99 |
|  | Parameter 1, $\beta_o$ | Parameter 2, $\beta_1$ | Parameter 5, CS+ habituation | Parameter 4, CS- habituation |  |  |  |
| **13** | **0.90** | **0.96** |  |  |  |  |  |

*Appendix 1—table 2 Continued on next page*

*Appendix 1—table 2 Continued*

| Model | Parameter 1, CS+ learning Rate | Parameter 2, CS- learning Rate | Parameter 3, CS+ habituation | Parameter 4, CS- habituation |
|---|---|---|---|---|
| 14 | 0.81 | 0.90 | –0.03 | 0.33 |

**Appendix 1—table 3.** Correlations between the actual parameters and recovered parameters for each model for extinction data.

Models in bold met the criteria for model comparison (all correlation values for all parameters > 0.20).

| Model | Parameter 1, CS+ learning rate | Parameter 2, CS- learning rate | Parameter 3, CS+ habituation | Parameter 4, CS- habituation |
|---|---|---|---|---|
| **1** | **0.59** | **0.45** | | |
| **2 (inertia)** | **0.51** | **0.46** | | |
| **3 (Bayesian)** | **0.83** | **0.69** | | |
| 4 | 0.28 | 0.07 | 0.46 | 0.93 |
| **5 (inertia + Bayesian)** | **0.80** | **0.82** | | |
| 6 (inertia) | 0.49 | 0.27 | 0.05 | -0.04 |
| 7 (Bayesian) | 0.32 | 0.40 | 0.05 | 0.40 |
| 8 (inertia + Bayesian) | 0.22 | 0.25 | -0.003 | 0.01 |

| Model | Parameter 1, CS+ learning rate | Parameter 2, CS- learning rate | Parameter 3, CS+ learning rate update | Parameter 4, CS- learning rate update | Parameter 5, regression $\beta_o$ | Parameter 6, regression $\beta_1$ | Parameter 7, regression $\beta_2$ |
|---|---|---|---|---|---|---|---|
| 9 | 0.47 | 0.33 | 0.11 | 0.01 | | | |
| 10 | 0.28 | 0.01 | 0.01 | 0.03 | 0.82 | 0.19 | |
| 11 | 0.85 | 0.11 | 0.99 | 0.86 | –0.31 | –0.03 | |
| 12 | 0.99 | -0.002 | 0.99 | 0.93 | 0.90 | 0.90 | –0.01 |

Indeed, most prior work that finds this model to work well has included substantially more trials and contingency reversals; here, we focus only on initial, rapid acquisition as opposed to change in threat value over many trials. Thus, when modeling contingencies and their change over longer durations, the hybrid model may provide an optimal fit, whereas initial acquisition of contingencies might reflect a specific case in which other models perform better.

Model recovery was performed by simulating data from the nine base models using the parameters from the participants best fit by each model. However, due to the unequal distribution of participants best fit by each model, we calculated the mean and standard deviation of the parameters from the best fit participants, and then sampled from the corresponding Gaussian distributions to create a total of 100 sets of parameters for each model. These parameters were then used to generate synthetic datasets. That simulated data was then fit with all the models to determine the extent to which the model selection approach would identify the model used to generate the data. Note that in some cases, for example in habituation models, we could sample very small parameter values for the habituation term, which would lead to selection of a simpler model. See confusion matrix for model recovery results (*Figure 2—figure supplement 2*).

Overall, the models were recovered reasonably well, except for models 6 and 9. It is important to note that models 1-8 are variants of a general model that captures the process of interest. We did not intentionally bias our selection of parameters to the parameter range that would best separate these models and improve model recovery. Thus, we chose the participants best fit by the models as a reasonable starting point for parameter and model recovery procedures. In this context, models 3 and 7 recovered relatively better, while model 9 (RW-PH) and model 6 did not recover as well.

*Figure 3—figure supplement 1* depicts empirical and modeled data overlays for CS- and CS+ during extinction.

### Additional analyses

In this report, we did not bin the participants into categories of anxiety (patients vs. healthy comparisons) or age (youth vs. adults). This is because both anxiety and age are naturally continuous variables, and it is not clear that breaking them down by categories offers an advantage, either empirically or conceptually. Nevertheless, when participants are binned categorically, we do not see a clear relationship between which model is picked and anxiety, as can be seen in *Figure 2—figure supplement 3*.

In addition to the primary analyses reported in the main text, we also examined associations between learning parameters and anxiety separately among youth participants and among adult participants, as described below.

### Conditioning

Anxiety was not significantly associated with CS+ threat conditioning or habituation rates in the youth group, $\beta s \leq 0.11$, $ps \geq 0.665$, and in the adult group, $\beta s \leq 0.53$, $ps \geq 0.289$. Anxiety severity was not significantly associated with CS- generalization rate in the youth group, $\beta s \leq 0.51$, $ps \geq 0.297$, and in the adult group, $\beta s \leq 0.45$, $ps \geq 0.297$. CS- safety learning rate was negatively correlated with anxiety in the youth group, $\beta = -0.27$, $p = 0.002$, as in the full sample, and marginally so in the adult group, $\beta = -0.24$, $p = 0.056$, suggesting an attenuation of anxiety-safety learning association with age. No other effects were observed.

Effects of brain structure emerged in the youth group. These included a negative association between brain stem volume and CS- safety learning rate, as well as bilateral accumbens moderation effects similar to those reported in the main text, $p_{FWE}s < 0.05$. Right amygdala showed a similar moderation effect at trend level, $p_{FWE} = 0.0628$. In the youth group, left accumbens showed a moderation effect similar to that reported in the main text and in the youth group, but at trend level, $p_{FWE} = 0.073$.

### Extinction

In the youth group and the adult group, anxiety was not significantly associated with CS + extinction rate, $\beta = 0.12$, $p = 0.227$, and $\beta = -0.16$, $p = 0.204$, respectively. CS- extinction rate was negatively correlated with anxiety in the youth group at comparable magnitude to what was observed across the full sample, $\beta = -0.20$, $p = 0.032$, but not in the adult group, $\beta = -0.09$, $p = 0.504$, suggesting an attenuation of the association between anxiety and CS- extinction with age. No other effects were observed.

In the youth group, no effects of brain structure emerged. In contrast, in the adult group, a significant positive association between right amygdala volume and CS+ extinction rate was noted, $p_{FWE} = 0.008$. Brain stem volume moderated the association between anxiety and CS+ extinction, $p_{FWE} = 0.012$. Anxiety-CS- extinction rate was moderated by left accumbens and brain stem volume at trend level, $p_{FWE}s < 0.073$.

### Quadratic effects of age

Some evidence suggests quadratic effects of age on threat learning, and extinction learning in particular (*Casey et al., 2015*; *Pattwell et al., 2012*). Exploratory analyses testing a quadratic effect of age instead of a linear effect did not yield significant effects in all analyses, $ps > 0.06$.

