## [Editor Report]

The authors present an investigation into the role of threat learning processes in symptoms of anxiety across a broad sample of subjects with and without clinical anxiety, across multiple age groups. Authors demonstrated weaker safety learning in those who were more anxious.

---

## [Decision Letter]

**Decision letter after peer review:**

Thank you for submitting your article "Computational Modeling of Threat Learning Reveals Links with Anxiety and Neuroanatomy" for consideration by *eLife*. Your article has been reviewed by 3 peer reviewers, and the evaluation has been overseen by Drs. Shackman (Reviewing Editor) and Michael Frank (Senior Editor).

Summary:

The authors present an investigation into the role of threat learning processes in symptoms of anxiety across a broad sample of subjects with and without clinical anxiety, across multiple age groups. The study uses a simple fear conditioning paradigm in which subjects first undergo an acquisition procedure to learn associations between visual CSs and loud noise USs, before undergoing an extinction procedure where the CS+ is presented without the US. Reinforcement learning models are fit to skin conductance responses, and parameters from these models are used to predict symptoms of anxiety. This demonstrated weaker safety learning (i.e. learning that the CS– did not predict threat) in those who were more anxious, and slower extinction in more anxious individuals, a result that was moderated by gray matter volume in a number of subcortical brain regions.

The reviewers expressed enthusiasm for the manuscript:

– This is an important area, and the results will be of interest to readers across the field of computational psychiatry, and mental health research more widely.

– This work has the potential to be an important study in the field of computational psychiatry. Investigations of threat learning processes in individuals with clinical anxiety are lacking, despite their clear relevance as a mechanism underlying symptoms of these disorders.

– [noted by all reviewers] The sample is large and includes subjects with clinically significant levels of anxiety who were not receiving medication.

– The modelling approach is carefully considered, and the authors have clearly been thorough in testing different model variants.

– The inclusion of brain volume data provides some interesting insights into how gray matter volume relates to these core learning processes.

– The authors explain the added value brought by the computational modelling approach well, and it clearly provides some interesting insights into the role of threat learning that would not be seen in more traditional analyses (e.g. prior work by this team).

– All published work on computational models of threat learning in SCR used comparably smaller samples with more limited age range (Li et al. 2011, Zhang et al. 2016, Tzovara et al. 2018, Homan et al. 2019). Hence, this is an important data set that could offer substantial insights into threat learning.

– This paper is of potential interest to researchers within the fields of clinical psychology, clinical neuroscience and computational psychiatry.

– Over recent years, both the use of dimensional measures of symptoms and adoption of computational modelling has advanced our understanding of psychopathology–related differences in cognitive and brain function. This has encompassed both original empirical studies and application of computational modelling to previously published datasets. Both these approaches have potential value and different advantages and disadvantages. The current study is an example of the latter approach.

– The consideration of nine alternate models in the modelling of SCR responses to the CS+ during acquisition is a strength of the paper.

– The application of computational modelling techniques to such a large differential fear conditioning dataset, with data from both healthy adult and child participants and unmedicated patients seeking treatment for anxiety is highly appealing.

Nevertheless, all 3 reviewers expressed some significant concerns, mostly centered on the analytic framework and other aspects of the approach. There was a consensus among us that the necessary revisions will entail considerable effort on the part of the authors.

– Modeling Approach

Concern: The modelling approach could be validated more robustly. For example, model and parameter recovery analyses are not presented, and so it is unclear how much confidence can be placed in the winning model and the parameter values derived from this model.

Recommendation: Include a more thorough validation of the modelling approach, adding model recovery and parameter recovery analyses. This would provide confidence in the conclusions drawn from the modelling by demonstrating that it is possible to recover the winning model and its parameters accurately.

Recommendation: It would be helpful to also show the BICs for different models to give the reader a clearer idea of how model fit differed across the different models.

Recommendation: It would be helpful if the authors could conduct Bayesian model comparison (see Stephan et al., Neuroimage, 2009) and report the population–level estimate of the proportion of participants best fit by each model. Given the age by anxiety analyses, it would also be helpful if they could report the best–fitting model for high and low anxious children (as assessed using the SCARED) and high and low anxious adults (as assessed using the STAI). Does the same model 'win' for both child and adult participants divided into low and high anxious subgroups?

Concern: Description of model estimation and comparison lacks detail. Model parameters are fit with fminsearch (using which starting values? random? grid search? just one starting value would hardly be sufficient. any parameter constraints?) using OLS minimisation. How is BIC then computed? How are the models compared – the methods mention "per participant and across the sample"; do you assume a fixed or random model structure? What is the purpose of a t–test (p. 13) in the model comparison?

Recommendation: Provide modeling details.

Concern: The model space of the acquisition session does not include any of the "best" threat learning models from previously published work (see below for some specific examples). This precludes a direct comparison; more importantly it may also miss out on the best model in this sample. First, previous work has simultaneously modelled CS– and CS+ responses. This is appropriate (for all sessions), since a priori the agent does not know which is the "safety" cue, and will presumably use the same mechanisms to update value for both cues. The authors state that standard models cannot account for the large increase from first CS– trial (before first reinforced CS+) to next trial; while this is correct, the authors have a model to account for that, and more trivially one may just leave out the first CS– response. Secondly, the observation function (which is only specified in the legend of table 1 – this should appear in the text) maps SCR onto the value terms of all models. All previous SCR modelling work has found SCR is more closely related to associability or uncertainty, or a mixture of these with value.

Concern: The models considered are broadly suitable but comparison of model fit across both SCRs to the CS+ and CS– during acquisition and SCRs to the CS+ and CS– in extinction is missing, which is a limitation.

Recommendation: Include prior models in the model space. Models should simultaneously include CS+ and CS– responses. To this end, they may leave out responses before the first reinforced CS+ trial. Model space should include additional variants of models 9 that account for a mapping onto associability (as in Li et al. 2011, Zhang et al. 2016) or combined associability and value (as in Homan et al. 2019) and a variant of model 3 that accounts for a mixture of uncertainty and value (as in Tzovara et al. 2018).

Related Concern from Another Reviewer: The authors do not consider alternate models for the SCRs to the CS– during acquisition or for the SCRs to the CS+ during extinction and they do not model the SCRs to the CS– during extinction at all.

Recommendations: It would be helpful to know which model best fits all the acquisition data (to both the CS+ and CS–). Does the hybrid model still perform less well than the non–hybrid models? And does model 8 still outperform model 7? (in models 8 and 9, as in model 7, the CS– value could also be modelled as updating after each UCS delivery with model 8 now having 6 free parameters and model 9 4 free parameters.) Similarly, it would be useful to have both CS– and CS+ trials during extinction modelled. Given the raw data indicates the response to both the CS+ and CS– drop off during extinction, understanding if decreased responding to the CS+ during extinction represents active specific updating of the CS+ UCS contingency or a non–specific decreases in expectancy of the UCS after both the CS+ and CS– is clearly important.

– Multiple Comparisons [Multiple Reviewers]

Concern – There are many post–hoc tests on association of model parameters with brain results and participant traits. It is unclear how this was planned and performed – how did the authors correct for multiple comparison across parameters, traits, and experiment phases?

Recommendation: provide appropriate, principled correction for multiple comparisons to mitigate familywise α inflation

Recommendation: Please state how many tests were conducted and what correction was applied for example how many brain regions were investigated, was grey matter volume in each of these related to each of the five learning rate parameters (across acquisition, generalisation and extinction), and were the moderating effects of anxiety, age and anxiety by age considered in each case?

Recommendation: please be clear in defining what "counts" as a family of analyses

– Specificity of key results

Concern: The relationship between anxiety and safety learning (as indexed by the habituation parameter to the CS– during acquisition) is one of the main findings.

Concern: The novelty of this study depends both on the strength of the computational analysis and on the extent to which the results provide new insights into the relationship between child and adult anxiety and differential fear conditioning relative to the non–computational analyses reported in the authors' prior publication on this dataset (Abdend et al., 2020). The relationship between anxiety and safety learning (as indexed by the habituation parameter to the CS– during acquisition) could be one such novel insight – however here a direct comparison of the relationship between anxiety and this parameter versus the habituation parameter to the CS+ is needed to ensure this isn't simply a non–specific habituation effect. The same issue holds for the finding of a relationship between anxiety and rate of extinction to the CS+ – the authors need to show this holds over and above any relationship between anxiety and drop off in responsivity to the CS–.

Recommendation: Here, a comparison of the relative strength of relationship between anxiety and participants' habituation parameter values for the CS– versus that for the CS+ is needed to ensure this isn't simply a non–specific habituation effect. Similarly, the authors need to directly compare the relationship between anxiety and drop off in SCRs to the CS+ during extinction against the relationship between anxiety and drop off in SCRs to the CS– during extinction to be able to make any claim specific to extinction to the CS+. This also applies to the moderating effects of brain structure on this relationship.

– SCR Approach

Concern: SCR data are likely not optimal for model fitting at the subject level; SCR is noisy, and the task involves very few trials. This may result in poorly estimated parameters; however it is not clear from the presented analyses how much of a problem this is.

Recommendation – Given the noisiness of SCR data and the limited number of trials available, parameter estimation could likely be improved substantially by using hierarchical Bayesian parameter estimation methods.

Concern: The quantification of the conditioned response is in line with some previous work, but in fact there is a range of possible analysis choices, summarised in Lonsdorf et al. 2017.

Recommendation: Given this heterogeneity, it would make sense to use the most sensitive method. Bach et al. 2020 Behaviour Research and Therapy and Ojala and Bach 2020 Neurosci Biobeh Rev discuss the most sensitive method to quantify threat conditioning from skin conductance data, and Bach et al. 2020 Nat Hum Behav provide background.

– Missing SCR Data

[Multiple reviewers] Concern: More than 1/3 of the sample are excluded due to "excessive missing data", defined as >50% missing trials.

Recommendations: It is unclear why these data were missing. Please address the explanation and please discuss the potential consequences for inference. Do excluded subjects systematically differ from those included? It would be helpful if the authors could include a table showing a break–down of the characteristics of included and excluded participants by age, gender, diagnostic status and anxiety levels and if they could address in the discussion any associated limitations in particular in relation to the age by anxiety analyses reported.

Concern: It is unclear why missing data are inter/extrapolated?

Recommendation: Use MSE. The model fitting procedure should be happy with missing data points if MSE (rather than SSE) is used as objective function. This would be more appropriate than giving the model a data value that does not exist. Censor, rather than interpolate.

– Disparate Dimensional Measures of Anxiety [Multiple Reviewers]

Concern: The combination of multiple anxiety measures (SCARED, STAI) into a single anxiety measure may not be valid, as it's reasonable to assume these are measuring subtly different constructs (aside from anything else, the SCARED measures recent symptoms while the STAI is a trait measure).

Concern: The main analyses appear to combine measures of anxiety severity across age groups as the primary variable of interest (using the SCARED for youth and the STAI for adults), and in one analysis find an interaction between age and anxiety that is driven by the younger age group. This raises some concerns as these measures are not necessarily measuring the same construct, and may limit the interpretability of these results.

Concern: Children and adults show opposing direction relationships between anxiety and fear generalisation. It is difficult to interpret the anxiety by age results given different measures were used to assess anxiety in children and in adults. This could potentially account for the different relationship between anxiety and fear generalisation in under and over 18s (see Figure 3B). I am not convinced that the authors can simply merge scores from the SCARED (which averages symptom scores based on child and adult reports) and STAI (an adult self–report questionnaire) as if they came from the same measure.

Recommendation: Absent some sort of direct quantitative harmonization (as in the GWAS literature for N/NE) or additional evidence to motivate the Z–score approach, it may not be appropriate to combine or directly compare the younger (SCARED) and older (STAI) samples (i.e. more appropriate to keep them distinct and examine them separately)

– Potential nonlinear effects of age

Concern: Only linear effects of age are modelled.

Recommendation: explore potential nonlinear effects (given that many developmental effects are nonlinear).

[Editors' note: further revisions were suggested prior to acceptance, as described below.]

Thank you for submitting your revised manuscript – "Computational Modeling of Threat Learning Reveals Links with Anxiety and Neuroanatomy in Humans" – for consideration by *eLife*. Your revised article has been evaluated by Drs. Frank (Senior Editor) and Shackman (Reviewing/Handling Editor) with guidance and consultation from 2 of the 3 original Reviewers.

The manuscript has improved, with one Reviewer emphasizing that "the authors have done a good job of addressing many of the suggestions and the manuscript is definitely substantially improved as a result."

Nevertheless, there was agreement that the paper would benefit from some additional modeling.

One Reviewer wrote that, Abend et al. have addressed some, but not all of my previous concerns. The model space of the acquisition session does not include any of the "best" threat learning models from previous publications. This precludes a direct comparison; more importantly it may also miss out on the best model in this dataset.

The authors now model CS– and CS+ responses simultaneously. Yet they still do not compare their models to any of the those used in previous studies of conditioned SCR (Li et al. 2011, Zhang et al. 2016, Tzovara et al. 2018, Homan et al. 2019). They claim that their model 9 is the hybrid model used in three of these studies, but actually it is not, because the update works differently (presently they update only upon US delivery, and both CS+ and CS– from PE in CS+ trials, where as previous RW and hybrid models used the standard update on every trial, on signed PE, and with no cross–update). They qualitatively argue why the associability term in previous work does not fit their data, but in fact they should just quantitatively compare models that map value and/or associability onto SCR. Finally, they argue that Tzovara's model can only update CS+ or CS– expectation with one parameter, and modelling both CS types within the same model would lead to parameter inflaton, but in fact Tzovara's winning learning model is entirely parameter–free (only has parameters for the observation function), so this is inaccurate.

The authors need to implement these pre–existing models and show quantitatively that they do not fit the data, rather than on rely on qualitative and methodological arguments for why this is unnecessary. Based on their rebuttal, it seems that they had already implemented the models, so it should be relatively straightforward to make the necessary direct ("head–to–head") comparisons.

Along related lines, the other Reviewer noted that, [page 11] The authors argue that to adapt the Tzovara et al. model, a 5th parameter would need to be added, and that this would make it too complex. It would still be worth testing this properly [per the other reviewer's comments], and it is possible (although probably unlikely based on the provided evidence) that the model could fit well enough that it still wins despite the additional complexity–related penalisation.

The Reviewers and Editors also discussed hierarchical Bayesian approaches, with one reviewer noting that, [page 19] Reviewer suggestions related to hierarchical Bayesian parameter estimation do not seem to have been addressed in the revision. For clarity, this refers to the method of estimating reinforcement learning model parameters (see van Geen and Gerraty, 2021), rather than the SCR quantification, and there's a strong possibility it would improve parameter estimation and hence sensitivity. I appreciate that this could be represent a substantial re–analysis.

During the consultation session, the 2 Reviewers agreed that the hierarchical Bayesian approach would be nice, but is not strictly necessary for the revision, with one noting that, "I wouldn't want to force them to redo all the model fitting with a hierarchical Bayesian approach, it would likely provide more accurate estimates, but it would be a fairly big undertaking. Some acknowledgement that other model–fitting procedures exist and could be more appropriate would be nice, however (this probably also applies to their SCR quantification methods)".

And the other writing that, "I agree. This model fitting approach is a major step and may not work for some of the models."

In sum, the Bayesian approach should –– at minimum –– be noted in the Discussion.

[Editors' note: further revisions were suggested prior to acceptance, as described below.]

Thank you for resubmitting your work entitled "Computational Modeling of Threat Learning Reveals Links with Anxiety and Neuroanatomy in Humans" for further consideration by *eLife*. Your revised article has been evaluated by Drs. Shackman (Reviewing Editor) and Frank (Senior Editor).

We are very pleased to accept your paper for publication pending receipt of a revision that adequately addresses the remaining suggestions from Reviewer #2. Congratulations in advance!

*Reviewer #2:*

This revision of the paper by Abend et al. is significantly stronger by including a direct comparison of the current model space with previous "best" models. There are two substantive and a few presentation points remaining.

1. The Tzovara et al. model is implemented differently from the original paper. The authors use separate regression parameters for CS+ and CS– trials for the Tzovara et al. model (i.e. 4 parameters) and it is not clear to me why – Tzovara et al. themselves had used only 2 regression parameters across these two trials types. With 2 instead of 4 parameters, the model might fare better (even though the fit/RSS will of course become worse). Could the authors include the original 2–parameter Tzovara et al. model?

2. Perhaps relatedly, I've been wondering about the initial values of the various terms in the update equations. Can they be presented in the tables? I'm wondering, for example, whether they used the same values for α_0 and β_0 in the β distribution of the Tzovara et al. model, but other readers might wonder about other models. Of course, it can make a big difference how a model is "initialised".

3. The authors make an effort to explain the difference between models 1–9, and models 10–14, by using different symbols on the LHS of the equations. Perhaps it could be made more prominent that models 1–9 predict CS–related SCR from US–related SCR and thus need no regression to scale the predictions to the data, whereas models 10–14 predict CS–related SCR from an arbitrary reinforcement variable.

4. I'm still unsure about using t–tests to compare BICs across a group. I have never seen this. What is the statistical foundation for this approach? I would suggest adding the RFX–model comparison by Stephan et al. as had been suggested by two reviewers in the context of the initial manuscript version. It is implemented in a matlab function (spm_BMS), requires only the matrix of BIC values, and takes a few seconds to run.

---

## [Author Response]

Summary:[…]– Modeling ApproachConcern: The modelling approach could be validated more robustly. For example, model and parameter recovery analyses are not presented, and so it is unclear how much confidence can be placed in the winning model and the parameter values derived from this model.Recommendation: Include a more thorough validation of the modelling approach, adding model recovery and parameter recovery analyses. This would provide confidence in the conclusions drawn from the modelling by demonstrating that it is possible to recover the winning model and its parameters accurately.

Thank you for this suggestion. We have now carried out parameter recovery and model recovery, as specified below.

Parameter recovery:

To carry out parameter recovery, we pulled the parameters for each model from the subjects to whom it was best fit and used those parameters to generate synthetic data from the model. We then fit the original model to the synthetic data and compared the corresponding actual and “recovered” parameters using a correlation coefficient. In general, the parameters were well recovered, with the exception of model 9 (RW-Pearce-Hall hybrid). We believe that our task design, probing our specific processes of interest, is not well-suited to drive the flexible learning component of the RW-PH model. Indeed, most prior work that finds this model to work well has included substantially more trials and contingency reversals; here, we focus only on initial, rapid acquisition and extinction as opposed to change in threat value over many trials. The main text now directs the reader (p. 15) to the full description of this rationale in Appendix 1 (pp. 57-58): “We carried out parameter recovery by using the parameters for each model from the participants to whom it was best fit and used those parameters to generate synthetic data from the model. For each model, we then fit the original model to the synthetic data and compared the corresponding actual and “recovered” parameters using correlation coefficients; see Appendix 1-table 2. In general, parameters were recovered well, with the exception of model 9 (RW-PH hybrid). This may suggest that this model is not well-suited to capture rapid conditioning. Indeed, most prior work that finds this model to work well has included substantially more trials and contingency reversals; here, we focus only on initial, rapid acquisition as opposed to change in threat value over many trials. Thus, when modeling contingencies and their change over longer durations, the hybrid model may provide an optimal fit, whereas initial acquisition of contingencies might reflect a specific case in which other models perform better.”

The correlation between the actual parameters and recovered parameters for each model was calculated and placed in a table (Appendix 1-table 2):

Model recovery:

We now note in Appendix 1 (p. 59): “Model recovery was performed by simulating data from all the models using the parameters from the participants best fit by each model. However, due to the unequal distribution of participants best fit by each model, we calculated the mean and standard deviation of the parameters from the best fit participants, and then sampled from the corresponding Gaussian distributions to create a total of 100 sets of parameters for each model. These parameters were then used to generate synthetic datasets. That simulated data was then fit with all the models to determine the extent to which the model selection approach would identify the model used to generate the data. Note that in some cases, for example in habituation models, we could sample very small parameter values for the habituation term, which would lead to selection of a simpler model.”

We summarized this with a confusion matrix (added to Appendix 1 as Figure 2—figure supplement 2):

The models were recovered reasonably well, with the exception of model 6 and model 9. It is important to note that models 1-8 are related variants of a model that captures the process of interest, i.e., they are rather similar (as we now note in p. 59). As such, the models are generally comparable and the accuracy of parameter and model recovery lower and more similar relative to model comparison between very different models. Note that we did not intentionally bias our selection of parameters or data to get full coverage of the parameter or data space best represented by these models, which would separate the differences and seemingly improve model recovery. To do this, one would need a process that exploits what each model is capturing differently than all other models. Thus, we chose the subjects best fit by the models as a reasonable starting point for parameter and model recovery procedures. In this context, models 3 and 7 recovered relatively better, while model 9 (RW-PH hybrid) and model 6 did not recover as well. This is now described in full in Appendix 1 (p. 59).

Recommendation: It would be helpful to also show the BICs for different models to give the reader a clearer idea of how model fit differed across the different models.

Thank you for this recommendation. In Author response image 1 we present a bar plot of the average BIC values for each of the models fit to the acquisition data to show similarity. (We performed similar analyses for extinction; those results are included in a later reply to a reviewer requesting more modeling of extinction). (Author response image 1) shows that the average BIC values for the acquisition models all fall within a range of ~10 values, reflecting the similarity of the models to each other. This suggests that the different variants capture the acquisition process comparably well. Our response to the next recommendation provides more detail on BIC comparison.

**Author response image 1. sa2fig1:** 

Recommendation: It would be helpful if the authors could conduct Bayesian model comparison (see Stephan et al., Neuroimage, 2009) and report the population–level estimate of the proportion of participants best fit by each model. Given the age by anxiety analyses, it would also be helpful if they could report the best–fitting model for high and low anxious children (as assessed using the SCARED) and high and low anxious adults (as assessed using the STAI). Does the same model 'win' for both child and adult participants divided into low and high anxious subgroups?

Please see our response to the previous recommendation for a figure showing mean BIC by model. In Author response image 2 we present the fraction of participants best fit by each model, as suggested.

We now note in p. 14 how we compared the models: “We then compared the distributions of BIC values generated for each model across all subjects using *t*-tests. First, the BIC value for each model for each participant was computed. The differences between sets of BIC values (by model, for all participants) were used to run *t*-tests to determine whether there were statistically significant differences in the distributions of values for each of the models. We also examined the best model for each subject. In this case the BIC values were compared, and the model with the lowest BIC value for each subject was selected (without *t*-test).” We then note in pp. 18-19: “To characterize model fits at a population level, we first performed a repeated-measures ANOVA on BIC values (model as fixed effect, participant as random effect, BIC value as dependent variable). We found that BICs varied across models, *F*(1,8)=38.98, *p*<0.001. We further found that models 7 and 8 performed better than all other models based on post-hoc comparisons (ps<0.05, Bonferroni-corrected). These models did not, however, statistically differ (*p*=0.20). Since model 7 is a simpler model, we chose it as the optimal model for our data…”With regard to proportion of models providing best fit, we do not see large stand-outs. Rather, we see that subjects are spread across models, reflecting natural variability among the subtle variants. Our goal with model fitting was to find models which captured the skin conductance response dynamics well, rather than support assertions about which model captured a particular observed behavior (as is often common in studies doing model comparison). In our original analyses, we used only simple models and found that without a habituation term in the model, the models fit poorly, and the learning rates tended to absorb aspects of the learning curves that were not due to differences in learning, because these models had large bias. The main thing that comes from this figure is that the simplest model (model 1) does not seem to capture the skin conductance data well, while the models with additional parameters can provide a more reliable estimate of skin conductance response values. Because the BIC distributions did offer a reasonable way to identify an accurate model, we based our model selection most strongly on that.

We want to emphasize that for this manuscript, we did not bin the participants into categories of anxiety (high vs. low anxiety) or age (youth vs. adults). This is because both anxiety and age are naturally continuous variables, and it is not clear that breaking them down by categories offers an advantage, either empirically or conceptually. Nevertheless, when subjects are binned categorically, we do not see a clear relationship between which model is picked and anxiety. Author response image 3 shows the fractions of participants best fit by each model, broken down by group. We do report in Appendix 1 results for the primary analyses (associations between learning parameters and anxiety, and moderation by brain structure). This way, the reader has results for both the full sample and by age group.

**Author response image 3. sa2fig3:** 

Concern: Description of model estimation and comparison lacks detail. Model parameters are fit with fminsearch (using which starting values? random? grid search? just one starting value would hardly be sufficient. any parameter constraints?) using OLS minimisation. How is BIC then computed? How are the models compared – the methods mention "per participant and across the sample"; do you assume a fixed or random model structure? What is the purpose of a t–test (p. 13) in the model comparison?Recommendation: Provide modeling details.

We apologize for not clarifying. We added more information to the text to clarify this. The model parameters were fit with *fminsearch*. We used starting values between 0.1 and 0.8 for learning rate. The habituation parameters started at 0. This is now noted in pp. 13-14. In pilot exploration, we found that this approach gave good fits, with a reasonable amount of computation. There were no parameter constraints for the acquisition data fits, as estimates naturally fell in the range -1:+1. This is noted in pp. 13-14. For extinction, unconstrained fits with all 9 models (described in more detail later) led to extreme estimates for some participants in order to accommodate the data. When learning rates fall outside 1/-1, the RW model is unstable. As such, learning rates were constrained to -1:+1, which improved model fit. Still, ~14% of participants had estimated pinned at -1 or 1, and for four participants, models did not converge. This is now all noted in p. 15. Data for these were not used in analyses.

The parameter set associated with the maximum log-likelihood (same as the minimum negative log-likelihood) was selected out of 10 iterations across parameter starting values. We calculated the BIC using a standard Gaussian assumption on the residuals:

BIC=kln(n)+nln(variance) Where *k* is the number of parameters in the model (2 or 4), and *n* is the number of data points in the dataset that was fit (which corresponds to the number of trials in CS+ and CS- data for the given phase being fit). This is the BIC calculation under the assumption that model errors are i.i.d. and normally distributed (Priestley, M.B. (1981). Spectral Analysis and Time Series. Academic Press. ISBN 978-0-12-564922-3. (p 375)). This is now noted in p.14.

As noted in our response to the previous recommendation, we also added information on how models were statistically compared (p. 14; pp. 18-19).

Concern: The model space of the acquisition session does not include any of the "best" threat learning models from previously published work (see below for some specific examples). This precludes a direct comparison; more importantly it may also miss out on the best model in this sample. First, previous work has simultaneously modelled CS– and CS+ responses. This is appropriate (for all sessions), since a priori the agent does not know which is the "safety" cue, and will presumably use the same mechanisms to update value for both cues. The authors state that standard models cannot account for the large increase from first CS– trial (before first reinforced CS+) to next trial; while this is correct, the authors have a model to account for that, and more trivially one may just leave out the first CS– response. Secondly, the observation function (which is only specified in the legend of table 1 – this should appear in the text) maps SCR onto the value terms of all models. All previous SCR modelling work has found SCR is more closely related to associability or uncertainty, or a mixture of these with value.Concern: The models considered are broadly suitable but comparison of model fit across both SCRs to the CS+ and CS– during acquisition and SCRs to the CS+ and CS– in extinction is missing, which is a limitation.Recommendation: Include prior models in the model space. Models should simultaneously include CS+ and CS– responses. To this end, they may leave out responses before the first reinforced CS+ trial. Model space should include additional variants of models 9 that account for a mapping onto associability (as in Li et al. 2011, Zhang et al. 2016) or combined associability and value (as in Homan et al. 2019) and a variant of model 3 that accounts for a mixture of uncertainty and value (as in Tzovara et al. 2018).

As requested by the reviewers, the 9 models were entirely refit to the combined CS+ and CS- trials during conditioning (see Figure 2—figure supplement 1), although it should be noted that it is not clear that threat and safety learning rely on the same circuitry and thus the same learning processes. We now clarify in the text in multiple places that both CSs were modeled in the same model (p. 10-15).

This was also done for the extinction process (discussed later; Figure 3—figure supplement 1). The models referenced in Li et al., Zhang et al., and Homan et al., are all the same version of the Pearce-Hall model which has already been used in our study (model 9). We apologize for not describing this model clearly. We include two associability terms in the model that drive changes in learning rate (for fitting CS+ and CS-), as now noted in p. 14.

We stress that the family of models chosen for this study was intentionally similar, and we are using them to improve the quality of the metrics (parameters) used to link the SCR data to the symptom and neural data. We note this in pp. 9-10.

We reviewed the variants of the PH model mentioned by reviewers (Li et al. 2011, Zhang et al. 2016, Homan et al. 2019) and examined the associability terms for model 9. We plotted the median values of the two associability parameters (α positive and α negative for CS+ and CS-), resulting from the model 9 fit (see Author response image 4) and compared the profiles to the acquisition data. However, the time course of associability for acquisition lacked the learning dynamics we observed in the SCR data, and for extinction, associability and value were highly similar.

**Author response image 4. sa2fig4:** 

As suggested, we also attempted to adapt the Tzovara et al. model, which uses uncertainty and value to co-determine the model’s predictions, to our reinforcement learning data. As Tzovara et al. conceived the model, it contains two parameters, and can only account for one type of reinforced data (i.e., CS+ or CS-, but not both). To model the response to both the CS+ and CS- simultaneously, we would need to add a third parameter (in addition to another two for the CS-) to the model, for a total of 5 parameters (more than any of our models, which have a maximum of 4 parameters). While possible, the BIC calculation would deduct points for this additional parameter, thus making it not an ideal model for these data. Author response image 5 comparing the Tzovara et al. uncertainty model to the CS+ data shows that the model fit across subjects does not pick up on the initial acquisition or subsequent habituation to the CS+:

**Author response image 5. sa2fig5:** 

Thus, it is possible that the process captured by Tzovara et al., which is modeled over many trials, is different than the rapid, initial acquisition process targeted in our study. Given this result and the difference in processes of interest, we did not pursue this model further. We provide this rationale in full in Appendix 1 (p. 60).Of note, we also discuss the absence of findings regarding the hybrid model in the context of this specific task (Discussion, p. 25): “Of note, prior work identified the RW-PH “hybrid” model as providing optimal fit to data for the expression of threat contingencies among healthy adults as well as adults with PTSD^12,33,34^. Here, we found that this model did not provide the best fit for rapid threat conditioning or extinction among the models tested. This discrepancy could potentially be attributed to the nature of threat learning processes studied. While we focused on rapid acquisition/extinction of threat contingencies which takes place over a short training schedule, prior modeling studies examined threat contingencies over much longer durations (over 80 CS+ trials). The hybrid model includes a cumulative associability term which could be more sensitive to tracking contingency values or the expression of conditioned fear as it is optimized over longer durations. Thus, differences in rapid, crude acquisition vs optimized expression of threat contingencies processes could account for model differences. As such, the current and prior work could be seen as complementary in terms of threat learning processes studied, and both sets of findings could usefully inform study design for future work.”

Related Concern from Another Reviewer: The authors do not consider alternate models for the SCRs to the CS– during acquisition or for the SCRs to the CS+ during extinction and they do not model the SCRs to the CS– during extinction at all.Recommendations: It would be helpful to know which model best fits all the acquisition data (to both the CS+ and CS–). Does the hybrid model still perform less well than the non–hybrid models? And does model 8 still outperform model 7? (in models 8 and 9, as in model 7, the CS– value could also be modelled as updating after each UCS delivery with model 8 now having 6 free parameters and model 9 4 free parameters.) Similarly, it would be useful to have both CS– and CS+ trials during extinction modelled. Given the raw data indicates the response to both the CS+ and CS– drop off during extinction, understanding if decreased responding to the CS+ during extinction represents active specific updating of the CS+ UCS contingency or a non–specific decreases in expectancy of the UCS after both the CS+ and CS– is clearly important.

Joint fitting of CS+ and CS- during acquisition:

As noted, we re-fit our nine models to both the CS+ and CS- during acquisition and performed model comparison as suggested. This is detailed in the previous responses and in the text. The hybrid model (model 9) did not perform particularly well, as can be seen in the bar chart of average BIC values across the models or fraction of best fit across the sample (Figure 2A). Notably, model 9 is also one of the two models (the other being model 6) that did not do well in model recovery. This model is designed to allow flexibility in learning rate, which is important in situations where there are changes in uncertainty over time. Because our task is a straightforward rapid acquisition/extinction task, we believe it might not necessarily provide the right statistics to lead to it fitting best. We note in this in Appendix 1 (pp. 59-60).

We performed a repeated-measures ANOVA on BIC values (model as fixed effect, subject as random effect), yielding a significant effect of model, *F*(1,8)=38.98, *p*<0.001, as noted in pp.18-19. Pairwise t-tests were conducted for model 9 versus all other models (Bonferroni corrected). Models 3, 4, 7, and 8 were significantly better than model 9, model 2 performed worse. Thus, we conclude that fitting the data from this task with this model does not produce parameter values that are optimal representations for this type of learning (i.e., rapid acquisition/extinction). This is now reported in Appendix 1 (pp. 59-60). Models 7 and 8, on the other hand, still perform best compared to all the other models. When directly compared to each other, model 8 and model 7 do not statistically differ (*p*=0.20). As before, since model 7 is a simpler model and they do not statistically differ, we chose model 7 as the optimal model for our data and used these parameters in subsequent analyses (pp.19-20). Thus, we have completely redone the analyses to account for these changes (pp. 19-21).

Extended modeling of extinction:

All 9 models were next fit to the CS+ and CS- extinction data; see Figure 3—figure supplement 1.

As noted in our response to previous recommendations, we provide information on joint CS-/CS+ fitting in p. 10-11. Several models had relatively low BIC values across the population, although overall there were no large differences between models, as can be seen Author response image 6.

**Author response image 6. sa2fig6:** Model fit: BIC values and fraction of best fit.

We now note in p. 20 that we conducted a mixed-effects ANOVA on the BIC values (subject as random effect, model as fixed effect) to determine whether any of the models were significantly different than other models, and found a difference between models, *F*(1,8)=26.89, *p*<0.001. As can be seen, this effect is driven primarily by worse fits of models 4 and 9. We then ran follow-up pairwise *t*-tests for model 3, which was picked most often as the best at the individual subject level, and found that model 3 was not statistically different than models 1,2,5,6,7, or 8. Since model 3 was the simplest among the better models, and picked most often for individual subject fits, we used its parameters to model extinction. This is all now described fully in the text (p. 20).

We have accordingly redone the entire relevant section of the results concerning the relationship between the extinction data, anxiety symptoms, and neural data using model 3 parameters (pp. 20-21).

– Multiple Comparisons [Multiple Reviewers]Concern – There are many post–hoc tests on association of model parameters with brain results and participant traits. It is unclear how this was planned and performed – how did the authors correct for multiple comparison across parameters, traits, and experiment phases?Recommendation: provide appropriate, principled correction for multiple comparisons to mitigate familywise α inflation

We apologize for not making this part sufficiently clear, and we appreciate the opportunity to clarify our analysis plan as well as test level (α) adjustments. We now specify in the Methods (p. 16) that for each of the two phases (conditioning, extinction), we first identified the model best accounting for the phase data (which are now modeled using both CS- and CS+). For each estimated parameter in the winning model, we then examined whether it was associated with anxiety severity, and the moderation of this association by age using a regression model. Conditioning effects were best modeled using a model with four parameters (2 for CS+ and 2 for CS- responses, as we show in the Results section), and thus the test level was determined via Bonferroni as α=0.05/(4 parameters)=0.0125. Extinction effects were best modeled using two parameters (one for each CS, see Results), and thus significance level for each tested effect was determined as α=0.05/(2 parameters)=0.025. In terms of imaging analyses, we examined whether brain vertex (cortical thickness)/subcortical structure (GMV) was associated with each learning parameter as moderated by age. All imaging analyses used FWE rate correction for multiple comparisons of α<0.05, whereby the family of tests for each analysis consisted of all vertices/subcortical structures across all tested effects. This is now all detailed in the Methods section, p. 16.

Recommendation: Please state how many tests were conducted and what correction was applied for example how many brain regions were investigated, was grey matter volume in each of these related to each of the five learning rate parameters (across acquisition, generalisation and extinction), and were the moderating effects of anxiety, age and anxiety by age considered in each case?Recommendation: please be clear in defining what "counts" as a family of analyses

We hope that our previous response clarifies the types of multiple-comparison correction used in analyses and what the tested effects were. Specifically, we note (p. 16) what counts as a family for behavioral analyses (all tested effects) and for imaging analyses (all vertices/subcortical structures across tested effects). We also now specify the number of vertices tested in the whole-brain cortical thickness analysis and moved the listing of subcortical structures tested for GMV from Appendix 1 to the main text (p. 15).

– Specificity of key resultsConcern: The relationship between anxiety and safety learning (as indexed by the habituation parameter to the CS– during acquisition) is one of the main findings.Concern: The novelty of this study depends both on the strength of the computational analysis and on the extent to which the results provide new insights into the relationship between child and adult anxiety and differential fear conditioning relative to the non–computational analyses reported in the authors' prior publication on this dataset (Abdend et al., 2020). The relationship between anxiety and safety learning (as indexed by the habituation parameter to the CS– during acquisition) could be one such novel insight – however here a direct comparison of the relationship between anxiety and this parameter versus the habituation parameter to the CS+ is needed to ensure this isn't simply a non–specific habituation effect. The same issue holds for the finding of a relationship between anxiety and rate of extinction to the CS+ – the authors need to show this holds over and above any relationship between anxiety and drop off in responsivity to the CS–.Recommendation: Here, a comparison of the relative strength of relationship between anxiety and participants' habituation parameter values for the CS– versus that for the CS+ is needed to ensure this isn't simply a non–specific habituation effect. Similarly, the authors need to directly compare the relationship between anxiety and drop off in SCRs to the CS+ during extinction against the relationship between anxiety and drop off in SCRs to the CS– during extinction to be able to make any claim specific to extinction to the CS+. This also applies to the moderating effects of brain structure on this relationship.

Thank you for this recommendation, which allowed us to examine stronger inferences regarding the specificity of effects. The association between anxiety and CS- habituation rate during conditioning remained significant when controlling for CS+ habituation rate, β=-0.239, *p*=0.001, indicating the specificity of anxiety effects to safety learning. This is now noted in p. 19-20, and in the Discussion, pp. 24-25. Similarly, the association between anxiety and CS- extinction rate maintained its magnitude when controlling for CS+ extinction rate, *p*=0.031, and also when controlling for CS- learning and habituation rates during conditioning, *p*=0.030, providing support for the specificity of the anxiety-extinction association (reported in p. 21). Likewise, the significant brain structure finding for CS- habituation and anxiety moderation by accumbens GMV remained significant when controlling for CS+ habituation rate *p*=0.040 (p. 20). The effect for extinction was not re-assessed as it was only at trend level.

– SCR ApproachConcern: SCR data are likely not optimal for model fitting at the subject level; SCR is noisy, and the task involves very few trials. This may result in poorly estimated parameters; however it is not clear from the presented analyses how much of a problem this is.Recommendation – Given the noisiness of SCR data and the limited number of trials available, parameter estimation could likely be improved substantially by using hierarchical Bayesian parameter estimation methods.

We agree that SCR data are indeed inherently noisy. This was a major motivation to accumulate a large sample of participants, especially in the studied populations. Accordingly, data collection took several years, as recruitment of the target populations (medication-free, clinically-diagnosed anxiety patients, including children, who are willing to take part in an aversive experiment) is a difficult, expensive, and time-consuming process. We opted to retain our original equipment and software during this period. Since it has been several years since data were collected and analyzed, and as our group has moved on to newer equipment for data collection and analysis since then, we, unfortunately, no longer have access to the original raw data and cannot reanalyze them. While model fit to the SCR data was good (as was model recovery) and the sample size was large, we are aware that not using newer methods could possibly decrease statistical power, rendering our approach rather conservative. It should also be noted, however, that recent work does not clearly indicate whether newer processing methods provide stronger or more valid estimates in the context of the paradigm used here (Kuhn, M., Gerlicher, A., and Lonsdorf, T. B. (2021, April 30). Navigating the manifold of skin conductance response quantification approaches – a direct comparison of Trough-to-Peak, Baseline-correction and model-based approaches in Ledalab and PsPM. https://doi.org/10.31234/osf.io/9h2kd). Nevertheless, we now note this as a limitation of the study in p. 28, citing papers by Bach et al. which provides background and examples for such methods: “Fourth, SCR data were analyzed using conservative methods (base to peak amplitude); future studies should consider utilizing more novel analysis methods which may increase sensitivity and statistical power^96-99^”.

Concern: The quantification of the conditioned response is in line with some previous work, but in fact there is a range of possible analysis choices, summarised in Lonsdorf et al. 2017.Recommendation: Given this heterogeneity, it would make sense to use the most sensitive method. Bach et al. 2020 Behaviour Research and Therapy and Ojala and Bach 2020 Neurosci Biobeh Rev discuss the most sensitive method to quantify threat conditioning from skin conductance data, and Bach et al. 2020 Nat Hum Behav provide background.

We appreciate the suggestion to re-analyze the dataset using alternative methods of analysis as indeed multiple different choices do exist. It is quite possible that changing the way that we analyze data in our group to a different method might increase the effects sizes, although, as noted above, it is not clear that this effect would be achieved (see reference to Kuhn et al.; newer methods might not necessarily show more sensitivity than the method used here). In any event, re-analyzing the data will likely change the nature of findings, from learning models to their associations with symptoms/brain structure, since different methods might not necessarily capture the same physiological processes (e.g., assuming a canonical phasic response vs possible changes in SC that include phasic but also tonic changes – our own recent work, under review elsewhere, supports the latter notion). As such, it is difficult for us to conclusively say that all alternative methods measure the same exact physiological process, and we cannot conclusively say (given recent evidence) which method is the ideal one in this context. In this case, the dataset should be re-analyzed using multiple methods, and we worry that having to analyze the data using multiple methods and indices might make interpretation difficult and the paper more challenging to read. Importantly, as we note in our response to the previous comments, we were also unable to reanalyze the data in this specific case, for practical reasons. Nevertheless, we agree that using the current method, while standard, might not necessarily be the most powerful method and could therefore be considered more conservative. As such, as noted above, we added an important limitation that pertains to the analysis method that also encourages future work to explore newer analytic methods (p. 28; see above). We cite the papers suggested by the Editor, as well as additional work by Bach et al., to provide context.

– Missing SCR Data[Multiple reviewers] Concern: More than 1/3 of the sample are excluded due to "excessive missing data", defined as >50% missing trials.Recommendations: It is unclear why these data were missing. Please address the explanation and please discuss the potential consequences for inference. Do excluded subjects systematically differ from those included? It would be helpful if the authors could include a table showing a break–down of the characteristics of included and excluded participants by age, gender, diagnostic status and anxiety levels and if they could address in the discussion any associated limitations in particular in relation to the age by anxiety analyses reported.

Indeed, a substantial amount of SCR was missing. We agree that this is an important point to clarify and discuss. We report the criteria and exclusion process, as recommended by Lonsdorf et al. (2019, *eLife*), in Appendix 1 (pp. 52-53). Exclusion was due to individuals with no response or with outlier values. A comparably large proportion of non-responders has previously been reported (e.g., Homan et al., 2019; Lonsdorf et al., 2019). To assess whether the SCR processing method led to this dropout level, we analyzed using newer approaches (as suggested above) a different, ongoing dataset of adolescents who completed this task. We observed ~25% missing data using similar criteria, suggesting that it is not the processing method. Instead, based on our extensive experience, the noisiness and difficulty in collecting good signal reflects the challenges of physiological data collection particularly across a wide age range and in those diagnosed with anxiety disorders, in the context of uninstructed aversive tasks. Due to these challenges, we chose conservative criteria, resulting in a large proportion of exclusion, since modeling was based on trial-by-trial data, and we needed sufficient usable trial-level data. Additionally, data collection for this study took place over several years, due to the need to acquire a large sample of patients across age; newer systems used today may be more sensitive, signal-wise.

As recommended by the Editor, we added Appendix 1-table 1 which details differences between included and excluded participants in terms of age, sex, anxiety (both diagnostic status and continuous severity), and IQ. We noted significant differences in age and sex (but not anxiety or IQ), such that excluded (non-responder) individuals tended to be older (*M*=23.4 vs. 18.8 years) and with a greater proportion of females (69.1% vs. 54.0%). This is now noted in the main text (p. 8) and Appendix 1. Indeed, age and sex differences in responsivity have been previously noted, but inconsistently (e.g., Boucsein, et al., 2012, *Psychophysiology*; Bari, et al., 2020, *J Biol Phys*; Ganella, et al., 2017, *Front Hum Neuro*).

It should be noted that primary analyses included age in the models, and as such, anxiety effects are independent of age effects. We further examined the primary analyses using sex as a covariate (p. 53).

Finally, we added a paragraph discussing these important issues to Appendix 1 (p. 53): “This proportion of exclusion is similar to excluded data in prior work^12,67^. See Appendix 1-table 1 for demographic and clinical differences as a function of inclusion/exclusion. No significant differences in exclusion were noted for anxiety severity (by diagnosis or continuously) or IQ, but excluded relative to included participants had a higher mean age and a greater proportion of females. Note that age was included in tested models, and thus observed anxiety effects are independent of age effects; further, results did not change when using sex as a covariate. Based on our experience, we believe that this exclusion proportion reflects challenges of physiological data collection during uninstructed aversive tasks, particularly across a wide age range and in those diagnosed with anxiety disorders, in conjunction with conservative exclusion rules since modeling required sufficient trial-by-trial data. These considerations should be taken into account in future research, particularly in studies across a wide age range and in those that consider sex, and require sufficient trial-level data. Implementing newer analytic techniques and equipment, as well as use of more robust aversive stimuli, could potentially mitigate this issue.”

Given other comments, we no longer report on significant Age x Anxiety interactions (see response to later comment). As requested, to address this important issue, we added another limitation referring to age and sex differences (p. 28): “Finally, there were differences in sex and age between individuals included and excluded (non-responders) from analyses, as previously reported^102,103^; while these differences did not influence our findings, they could still limit generalizability and thus future studies should consider such potential differences.”

We hope that this information could help researchers by alerting them to such potential differences in responsivity.

Concern: It is unclear why missing data are inter/extrapolated?Recommendation: Use MSE. The model fitting procedure should be happy with missing data points if MSE (rather than SSE) is used as objective function. This would be more appropriate than giving the model a data value that does not exist. Censor, rather than interpolate.

We chose to impute missing time-series data points since model generation relied on trial-by-trial data for a limited number of trials, and learning effects were very rapid. We chose to assume that missing data points lie between the recorded data around them and impute missing data, rather than censor since we were worried that censoring would lead to spurious estimates. We now provide this rationale in Appendix 1, p. 52.

– Disparate Dimensional Measures of Anxiety [Multiple Reviewers]Concern: The combination of multiple anxiety measures (SCARED, STAI) into a single anxiety measure may not be valid, as it's reasonable to assume these are measuring subtly different constructs (aside from anything else, the SCARED measures recent symptoms while the STAI is a trait measure).Concern: The main analyses appear to combine measures of anxiety severity across age groups as the primary variable of interest (using the SCARED for youth and the STAI for adults), and in one analysis find an interaction between age and anxiety that is driven by the younger age group. This raises some concerns as these measures are not necessarily measuring the same construct, and may limit the interpretability of these results.Concern: Children and adults show opposing direction relationships between anxiety and fear generalisation. It is difficult to interpret the anxiety by age results given different measures were used to assess anxiety in children and in adults. This could potentially account for the different relationship between anxiety and fear generalisation in under and over 18s (see Figure 3B). I am not convinced that the authors can simply merge scores from the SCARED (which averages symptom scores based on child and adult reports) and STAI (an adult self–report questionnaire) as if they came from the same measure.Recommendation: Absent some sort of direct quantitative harmonization (as in the GWAS literature for N/NE) or additional evidence to motivate the Z–score approach, it may not be appropriate to combine or directly compare the younger (SCARED) and older (STAI) samples (i.e. more appropriate to keep them distinct and examine them separately)

Thank you for this interesting comment. We agree that the SCARED and STAI are not the same measure, despite each being considered a ‘gold standard’ for the overall assessment of anxiety severity for its age group, and thus may not be fully comparable. At the same time, given the developmental trajectory of anxiety (Beesdo et al., 2009, *Psych Clin N Am*) and the great interest in developmental effects on threat learning circuitry in the context of anxiety and its treatment (e.g., Casey, et al., 2015, *Neuron*; Shechner, et al., 2015, *Dep and Anx*; King, et al., 2014, *Stress*), which was the impetus for this study, we believe it is still important to report on age effects across the age continuum. Balancing the need to comprehensively report our results as intended with the potential incongruity of anxiety measures, we repeated the primary analyses separately in youth and adults while also keeping the original analyses.

To alert the reader to this important issue and to the fact that additional analyses were conducted, we provide this acknowledgement relatively early in the Methods section (p. 7): “Given that the SCARED and STAI might not capture the construct of anxiety in an identical manner, we also report on SCARED analyses in youth and STAI analyses in adult participants separately.”

We retained the analyses examining associations between anxiety and learning across the age continuum since these already control for age. Further, we now report in the Results section (p. 21) that additional analyses were performed, as well as the rationale, and provide a brief summary. We provide the results in full in Appendix 1. Specifically, the brief summary in the main text notes: “In addition to the primary analyses examining associations between anxiety and learning parameters and the moderation of these associations by brain structure across the full sample, we also examined these effects separately among youth and adult participants. These are reported in full in Appendix 1 (see below). Briefly, they suggest that observed anxiety effects on safety learning and extinction are attenuated with age. Further, these suggest moderation of anxiety-safety learning rate by several structures (bilateral nucleus accumbens, brain stem, and amygdala) in youth participants but not in adults. In contrast, extinction rates were associated with amygdala, accumbens, and brainstem volume moderation in adults, but not in youths; see Appendix 1.”

The full text in Appendix 1 (pp. 61-62) states: “In addition to the primary analyses reported in the main text, we also examined associations between learning parameters and anxiety separately among youth and adult participants, as described below.

Conditioning. Anxiety was not significantly associated with CS+ threat conditioning or habituation rates in the youth group, βs≤0.11, *p*s≥0.665, and in the adult group, βs≤0.53, *p*s≥0.289. Anxiety severity was not significantly associated with CS- generalization rate in the youth group, βs≤0.51, *p*s≥0.297, and in the adult group, βs≤0.45, *p*s≥0.297. CS- safety learning rate was negatively correlated with anxiety in the youth group, β=-0.27, *p*=0.002, as in the full sample, and marginally so in the adult group, β=-0.24, *p*=0.056, suggesting an attenuation of anxiety-safety learning association with age. No other effects were observed.

Effects of brain structure emerged in the youth group. These included a negative association between brain stem volume and CS- safety learning rate, as well as bilateral accumbens moderation effects similar to those reported in the main text, *p_FWE_*s<0.05. Right amygdala showed a similar moderation effect at trend level, *p_FWE_*=0.0628. In the youth group, left accumbens showed a moderation effect similar to that reported in the main text and in the youth group, but at trend level, *p_FWE_*=0.073.

Extinction. In the youth group and the adult group, anxiety was not significantly associated with CS+ extinction rate, β=0.12, *p=*0.227, and β=-0.16, *p*=0.204, respectively. CS- extinction rate was negatively correlated with anxiety in the youth group at comparable magnitude to what was observed across the full sample, β=-0.20, *p*=0.032, but not in the adult group, β=-0.09, *p*=0.504, suggesting an attenuation of the association between anxiety and CS- extinction with age. No other effects were observed.

In the youth group, no effects of brain structure emerged. In contrast, in the adult group, a significant positive association between right amygdala volume and CS+ extinction rate was noted, *p_FWE_*=0.008. Brain stem volume moderated the association between anxiety and CS+ extinction, *p_FWE_*=0.012. Anxiety-CS- extinction rate was moderated by left accumbens and brain stem volume at trend level, *p_FWE_*s<0.073.”

Importantly, we also added a full paragraph in the Discussion section in which the differences between analyses in full vs. age-specific samples are discussed (pp. 25-26). Specifically, the text reads: “While the design of this study offers a unique opportunity to examine age effects on threat learning as these relate to anxiety severity, an inherent challenge that arises in research on anxiety along the lifespan is how to combine anxiety data from youth and adult participants. Although the SCARED and STAI are each considered “gold standard” measures in their respective target populations, they nevertheless are not identical. Under the assumption that these capture similar constructs, we uncovered several anxiety effects across the full age range. Alternatively, one may consider these measures to be incomparable, in which case analyses are restricted to specific age groups. This alternative approach indicated an attenuation of anxiety- learning associations with age. As such, one interpretation of the discrepancies is that observed effects may in fact be instrument-specific. Another interpretation is that limited range and sample size reduced statistical power to detect associations in the different sub-groups. In either case, consideration of harmonizing clinical data is needed in future research that examines individuals across a wide age range.”

– Potential nonlinear effects of ageConcern: Only linear effects of age are modelled.Recommendation: explore potential nonlinear effects (given that many developmental effects are nonlinear).

We previously reported quadratic effects (all with null results) in Appendix 1. These results are reported in p. 62.

[Editors' note: further revisions were suggested prior to acceptance, as described below.]

The manuscript has improved, with one Reviewer emphasizing that "the authors have done a good job of addressing many of the suggestions and the manuscript is definitely substantially improved as a result."

Thank you for this feedback. We agree that subsequent recommendations in the second round of reviews have also significantly improved our manuscript and the clarity of our findings.

Nevertheless, there was agreement that the paper would benefit from some additional modeling.One Reviewer wrote that, Abend et al. have addressed some, but not all of my previous concerns. The model space of the acquisition session does not include any of the "best" threat learning models from previous publications. This precludes a direct comparison; more importantly it may also miss out on the best model in this dataset.The authors now model CS– and CS+ responses simultaneously. Yet they still do not compare their models to any of the those used in previous studies of conditioned SCR (Li et al. 2011, Zhang et al. 2016, Tzovara et al. 2018, Homan et al. 2019). They claim that their model 9 is the hybrid model used in three of these studies, but actually it is not, because the update works differently (presently they update only upon US delivery, and both CS+ and CS– from PE in CS+ trials, where as previous RW and hybrid models used the standard update on every trial, on signed PE, and with no cross–update). They qualitatively argue why the associability term in previous work does not fit their data, but in fact they should just quantitatively compare models that map value and/or associability onto SCR. Finally, they argue that Tzovara's model can only update CS+ or CS– expectation with one parameter, and modelling both CS types within the same model would lead to parameter inflaton, but in fact Tzovara's winning learning model is entirely parameter–free (only has parameters for the observation function), so this is inaccurate.The authors need to implement these pre–existing models and show quantitatively that they do not fit the data, rather than on rely on qualitative and methodological arguments for why this is unnecessary. Based on their rebuttal, it seems that they had already implemented the models, so it should be relatively straightforward to make the necessary direct ("head–to–head") comparisons.Along related lines, the other Reviewer noted that, [page 11] The authors argue that to adapt the Tzovara et al. model, a 5th parameter would need to be added, and that this would make it too complex. It would still be worth testing this properly [per the other reviewer's comments], and it is possible (although probably unlikely based on the provided evidence) that the model could fit well enough that it still wins despite the additional complexity–related penalisation.

These suggestions are valid. In response, we have fit 5 additional models to our data, including three from Li et al., 2011: Hybrid(V), Hybrid(α) and Hybrid(V+α) and the Tzovara et al., 2018 probabilistic uncertainty model with and without an additional habituation parameter. We fit these 5 models to our acquisition data and 3 of the models (Li et al., 2011) to the extinction data. It is not possible to fit the Tzovara model in a meaningful way to extinction because there are no UCSs. This led to a total of 14 models used to model acquisition data and 12 models used to model extinction data. We have updated the manuscript to reflect this (detailed below).

We found that on average, these models provided reasonable fits to subject data. Please see Figure 2—figure supplement 1 and Figure 3—figure supplement 1 in the Appendix for all model and data overlays that include the new models for acquisition and extinction. Although we were able to fit the models, the Li et al. models had a large number of parameters relative to the number of data points available to fit them. After fitting these models, we also carried out parameter recovery on all of the models to evaluate the quality of the fits. We found that the Li et al., models had poor parameter recovery. Therefore, to improve the model selection process, we imposed a minimal criterion on the correlation coefficients of the parameters to determine which models should be considered in final model comparison. We imposed a criterion that all parameter correlation coefficients for a given model should be ≥0.20, if we subsequently included that model in model comparison.

Many parameters could not be recovered for the new models (see Appendix 1- Table 2). The Li et al., models had considerable additional complexity, and thus it is likely that the regression step that adds 1-3 additional parameters was allowing for too much flexibility to fit noise in the data. Our original model 9, which we called the Pearce-Hall Hybrid model, lacked this regression step, but was otherwise a similar model. This model also did not survive parameter recovery, suggesting that this family of models was not the best set of models for this particular dataset and processes modeled, despite previous reports and use of these models for other datasets. Importantly, as we note in the text, this dataset features a limited number of trials per subject in order to capture rapid learning processes; while we compensate for that with a large number of subjects, perhaps those models are better suited for processes that use more trials for each subject to estimate the larger number of parameters.

The Tzovara models fit the data reasonably well, and because parameters could be recovered (Appendix table 1- table 2), were included in model comparison. However, these models did not produce an average BIC better than the previously chosen model (Figure 2A). Pairwise t-tests showed that models 7 and 8 remained the best models (*p*s<0.05, Bonferroni-corrected). As these two models did not differ from each other, we chose the simpler model (model 7, Bayesian learning rate and habituation) and the original conclusions from this section did not change.

Of note, we also tried another variant of the Tzovara model with two additional habituation parameters (habituation for CS+ and CS- for 6 total parameters) but the habituation parameters of this model were not recoverable during parameter recovery (p. 14; Appendix 1-table 2). Thus, we retained the model with a single habituation parameter.

Using the same methods, we evaluated the new models for extinction data, which included the 3 Li et al., models (Figure 3—figure supplement 1). After parameter recovery and imposing a criterion for parameter recovery, four models remained for model comparison: models 1, 2, 3 and 5 (Appendix 1- Table 3). These models were not statistically different from each other (*p*s>0.05, Bonferroni-corrected), and model 3 was the best fitting model for more subjects than the competing models (Figure 3B). Thus, we retained our original model and use of parameters from this model to correlate with anxiety severity and brain volume.

Please see updated figures and tables that account for the additional models:

– Figure 2A, 2B (main text): updated bar plots for models compared for acquisition

– Figure 2—figure supplement 1: model/data overlays for all 14 models fit to acquisition

– Figure 2—figure supplement 3: fractions of subjects by age/anxiety for acquisition

– Figure 3A, 3B (main text): updated bar plots for models compared for extinction Figure 3—figure supplement 1: model/data overlays for all 12 models fit to extinction

– Figure 3—figure supplement 1: model/data overlays for all 14 models fit to extinction

– Appendix 1-table 2: parameter recovery for acquisition

– Appendix 1-table 3: parameter recovery for extinction

The Reviewers and Editors also discussed hierarchical Bayesian approaches, with one reviewer noting that, [page 19] Reviewer suggestions related to hierarchical Bayesian parameter estimation do not seem to have been addressed in the revision. For clarity, this refers to the method of estimating reinforcement learning model parameters (see van Geen and Gerraty, 2021), rather than the SCR quantification, and there's a strong possibility it would improve parameter estimation and hence sensitivity. I appreciate that this could be represent a substantial re–analysis.During the consultation session, the 2 Reviewers agreed that the hierarchical Bayesian approach would be nice, but is not strictly necessary for the revision, with one noting that, "I wouldn't want to force them to redo all the model fitting with a hierarchical Bayesian approach, it would likely provide more accurate estimates, but it would be a fairly big undertaking. Some acknowledgement that other model–fitting procedures exist and could be more appropriate would be nice, however (this probably also applies to their SCR quantification methods)".And the other writing that, "I agree. This model fitting approach is a major step and may not work for some of the models."In sum, the Bayesian approach should –– at minimum –– be noted in the Discussion.

Thank you for this suggestion and we apologize for not acknowledging other model fitting techniques more clearly. We have updated the Discussion section to include a section acknowledging Bayesian parameter estimation and potential future work in this direction. We acknowledge the suggested papers in addition to others describing hierarchical Bayesian methods. Please see Discussion, pages 29,30:

“As a final technical comment, reinforcement learning model parameters were estimated using maximum likelihood techniques on individual subjects followed by model comparison. Future work could expand on this by using hierarchical Bayesian parameter estimation to reduce the variance around parameter estimates^105-107^. However, choosing prior distributions within the hierarchical Bayesian approach is not trivial and may not work for all of the models tested in this study. As such, future work could focus on fewer models, such as those that survived parameter recovery in this study, test a range of priors, and determine whether parameter estimation could be improved.”

We have also expanded our description of other SCR quantification methods. (Discussion, page 29):

“Fourth, SCR data were analyzed using a single method which, while established, relies only on directly-observable effects. Future studies may consider using novel, computational analysis methods which could potentially reveal effects not observed using the current method^96-99^. Along these lines, a multiverse approach may be used in future work to comprehensively compare multiple methods of quantifying threat learning”.

[Editors' note: further revisions were suggested prior to acceptance, as described below.]

We are very pleased to accept your paper for publication pending receipt of a revision that adequately addresses the remaining suggestions from Reviewer #2. Congratulations in advance!Reviewer #2 (Recommendations for the authors):This revision of the paper by Abend et al. is significantly stronger by including a direct comparison of the current model space with previous "best" models. There are two substantive and a few presentation points remaining.1. The Tzovara et al. model is implemented differently from the original paper. The authors use separate regression parameters for CS+ and CS– trials for the Tzovara et al. model (i.e. 4 parameters) and it is not clear to me why – Tzovara et al. themselves had used only 2 regression parameters across these two trials types. With 2 instead of 4 parameters, the model might fare better (even though the fit/RSS will of course become worse). Could the authors include the original 2–parameter Tzovara et al. model?

Thank you for this detail and for the suggestion. We have repeated the fitting of the Tzovara model with only 2 regression parameters for CS+ and CS- trials. Previously, we had tried to give the model the best chance of success by using separate regressors for CS+ and CS-, as the CS+ data has different dynamics than the CS- data. The model and data overlays for the Tzovara model with 2 parameters are shown in the updated Figure 2—figure supplement 1. For completeness, we also repeated the modeling using additional habituation terms for a Tzovara model with 4 total parameters. The model and data overlays are also shown in updated Figure 2—figure supplement 1. We replaced the previous models 13 and 14 with the requested versions of the Tzovara models in all aspects of the manuscript. Parameter recovery values have been updated in Appendix1-Table 2. These are the new models 13 and 14 to replace those reported previously. Model 13 survived the parameter recovery criteria, whereas model 14 did not. Thus, model 13 was included in the revised versions of Figure 2A and Figure 2—figure supplement 3 and BIC analyses.

We have also updated the uploaded code (available on a GitHub page) to reflect the most recent versions of the Tzovara models.

2. Perhaps relatedly, I've been wondering about the initial values of the various terms in the update equations. Can they be presented in the tables? I'm wondering, for example, whether they used the same values for α_0 and β_0 in the β distribution of the Tzovara et al. model, but other readers might wonder about other models. Of course, it can make a big difference how a model is "initialised".

Thank you for this suggestion. We have added the initialization values, for both conditioning and extinction models, to the table of equations (Table 1) for completeness and clarity.

3. The authors make an effort to explain the difference between models 1–9, and models 10–14, by using different symbols on the LHS of the equations. Perhaps it could be made more prominent that models 1–9 predict CS–related SCR from US–related SCR and thus need no regression to scale the predictions to the data, whereas models 10–14 predict CS–related SCR from an arbitrary reinforcement variable.

Thank you for this suggestion to improve the clarity of our modeling procedures. We have added the following text to the manuscript in the Methods on page 14:

“In summary, for models 1-9, predictions were not scaled to the data by using regression. Models 1-9 predict CS-related SCR from US-related SCR. For models 10-14, CS-related SCR is predicted using a regression to associability, value, or uncertainty, as described for each model. For models 1-9, outcome was modeled as a continuous variable. For models 10-14, we modeled outcome as a binary value of 0 or 1, as described in past publications using these models.”

4. I'm still unsure about using t–tests to compare BICs across a group. I have never seen this. What is the statistical foundation for this approach? I would suggest adding the RFX–model comparison by Stephan et al. as had been suggested by two reviewers in the context of the initial manuscript version. It is implemented in a matlab function (spm_BMS), requires only the matrix of BIC values, and takes a few seconds to run.

We apologize that our model comparison procedures were unclear. The referenced paper by Stephan et al., (2009), “Bayesian model selection for group studies,” states a description of the random effects procedure as was used in our manuscript:

“A straightforward random effects procedure to evaluate the between-subject consistency of evidence for one model relative to others is to use the log-evidences across subjects as the basis for a classical log-likelihood ratio statistic, testing the null hypothesis that no single model is better (in terms of their log-evidences) than any other. This essentially involves performing an ANOVA, using the log evidence as a summary statistic of model adequacy for each subject.

This ANOVA then compares the differences among models to the differences among subjects with a classical F-statistic. If this statistic is significant one can then compare the best model with the second best using a post hoc t-test.”

“The most general implementation would be a repeated-measures ANOVA, where the log-evidences for the different models represent the repeated measure. At its simplest, the comparison of just two models over subjects reduces to a simple paired t-test on the log-evidences.”

This was the methodology used in our manuscript as described in the Methods (pp. 15-16)­.

In addition, per the Reviewer’s suggestion, we have run the spm_BMS function from the SPM imaging toolbox on our BIC values. Using a flat prior across the models, we reproduced the posterior probabilities of the six models compared for the acquisition data:

Model 1: 0.0351

Model 4: 0.1356

Model 5: 0.1950

Model 7: 0.1423

Model 8: 0.2369

Model 13: 0.2551

These values are nearly identical to the fraction of subjects best fit by each model, shown in Figure 2A. Given that we do not expect that prior probabilities of each model should be anything but uniform, we did not expect significant differences between the RFX model comparison method and the random effects procedure.